# BEYOND WORST-CASE: DIMENSION-AWARE PRIVACY FOR BLACK-BOX GENERATIVE MODELS

## ABSTRACT

Black-box differentially private generative models often appears more private than worst-case accounting suggests, leaving a gap between formal Differential Privacy (DP) budgets and the observed weakness of membership inference attacks. We address this gap from a test-centric $f$-DP perspective. On the training side, we show that Differentially Private Stochastic Gradient Descent (DP–SGD) provides function-level stability, which can be quantified through loss-path kernels rather than parameter proximity. On the sampling side, the high-dimensional latent randomness of modern generators yields approximate Gaussian behavior, enabling a clean reduction to Gaussian DP. Combining these ingredients gives an effective signal parameter with small slack. The resulting envelopes predict that black-box distinguishability decreases with dataset size and effective latent dimension, and grows only sublinearly across multiple releases, while leaving formal DP budgets unchanged. Simulations and empirical tests confirm these predictions and align with observed attack performance, suggesting that our framework offers a practical and conservative tool for auditing the privacy of DP-trained generative models.

## 1 INTRODUCTION

As generative models become widespread, protecting training data with standards like Differential Privacy (DP) (Dwork, 2006) is more critical than ever. A widely adopted practice is training or fine-tuning with Differentially Private Stochastic Gradient Descent (DP-SGD) (Abadi et al., 2016), which adds calibrated noise to clipped per-example gradients. By post-processing, a model trained with DP-SGD inherits the DP guarantees regardless of how it is later used.

Most modern models are deployed as black boxes, served through APIs where attackers can only see the outputs. The model's parameters and training data are kept hidden. While this setup limits white-box attacks, it poses a new problem for privacy analysis: determining the practical privacy strength when an attacker cannot control the latent randomness used during generation.

Many studies on privacy-preserving generative models evaluate the utility-privacy tradeoff primarily through downstream quality metrics, often without a deeper analysis of the resulting privacy guarantees in practical settings. Examples range from PATE-style generators and GAN variants, as well as diffusion-based approaches and evaluation frameworks (Jordon et al., 2018; Chen et al., 2020; Harder et al., 2021; Vinaroz et al., 2022; Dockhorn et al., 2023; Greenewald et al., 2024). At the same time, empirical evidence (Annamalai et al., 2024) reports a striking black-box phenomenon: even with large accounting budgets (e.g., $\varepsilon = 4$), strong membership inference attacks can perform near random guessing. One view is that current attacks are still not strong enough. Alternatively, another view is that the effective privacy is stronger than the accounting suggests. Related theoretical hints exist in stylized linear models (Pierquin et al., 2025), where high-dimensional random inputs can reduce leakage in the $f$-DP sense.

Our view is test-based and does not violate the post-processing principle, where the formal DP parameters remain unchanged in the worst case, but the strength of black-box testing may drop sharply in real pipelines. We explain this gap within the $f$-DP framework (Dong et al., 2022). We focus on the canonical setting of DP-SGD training, which provides a clean and representative case for analysis. In practice, two factors affect the testing tradeoff curve as shown in Figure 1. First, DP-SGD induces *function-level stability*: generators trained on neighboring datasets are close

in a geometry tied to their training paths. Second, generation relies on *high-dimensional latent randomness* that the attacker does not control, which dilutes distinguishability at inference time.

To make this precise for modern networks, we use path kernels to connect training dynamics with function space. Unlike the static neural tangent kernel (NTK) (Jacot et al., 2018), path kernels aggregate gradient features along the optimization trajectory and better reflect finite-width learning. We work with *loss path kernel* (LPK) (Chen et al., 2023; 2025) to analyze the stability of DP-SGD, which has made significant progress in areas such as neural network generalization and neural architecture search. Leveraging Gaussianization tools on the latent input, we obtain hypothesis-testing guarantees that align with the empirical robustness reported in Annamalai et al. (2024). Although our analysis focuses on DP-SGD, we conjecture this amplification effect extends to other DP training mechanisms. Future work can continue with a similar analytical perspective to extend our analysis and conclusions to other DP generation mechanisms and their variants. We summarize our main contributions as follows.

- We leverage the hypothesis–testing ($f$-DP/GDP) perspective to analyze black-box use of DP-trained deep generative models, showing that distinguishability is jointly governed by DP–SGD stability and latent randomness rather than worst-case accounting alone.

- We establish function-level stability in the LPK geometry, providing an explicit $1/n$ rate under fixed subsampling and a direct bridge from training trajectories to output-level tests without requiring parameter proximity.

- By combining quantitative Gaussianization bounds with LPK stability, we derive GDP envelopes whose effective parameter decreases with dataset size and effective input dimension; the derivation isolates small slack $(\gamma_d, \omega)$ that are tracked empirically.

- We extend the analysis to multiple releases and validate the predictions on DP–SGD-trained VAEs, observing sublinear $\sqrt{m}$ composition of the effective signal and identifying regimes where slack accumulation limits amplification.

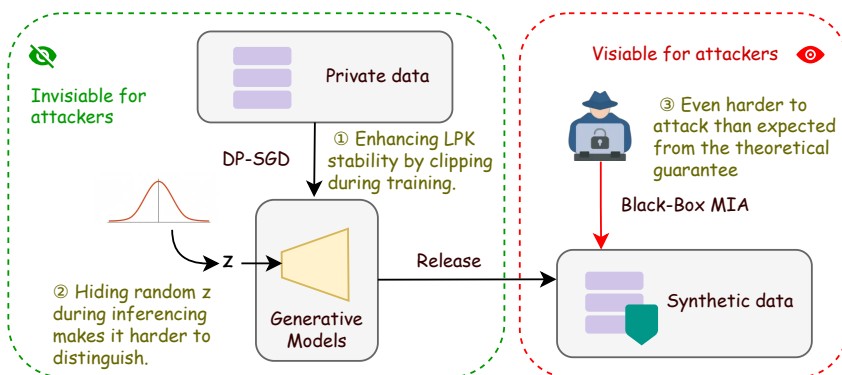

Figure 1: Generative models trained with DP-SGD are more difficult to attack in black-box settings.

## 2 RELATED WORK

### 2.1 DP GENERATIVE MODELS

Early differentially private synthetic data generation focused on estimating private statistics for tabular data (Zhang et al., 2016; McKenna et al., 2021). More recent efforts leverage expressive generative neural networks trained with differential privacy such as PATE (Papernot et al., 2017; Jordon et al., 2018; Long et al., 2021), DP-SGD (Torkzadehmahani et al., 2019; Dockhorn et al., 2023; Jiang & Sun, 2023) or other methods (Chen et al., 2020; Cao et al., 2021; Harder et al., 2021; Vinaroz et al., 2022; Jiang et al., 2023; Greenewald et al., 2024). Grounded in the post-processing theorem, these methods preserve the formal budget during sampling and can mitigate white-box and, more restrictively, black-box attacks. However, empirical studies report that black-box membership inference attacks are often far weaker than worst-case accounting suggests (Annamalai et al., 2024). Our work

does not propose a new generative algorithm; rather, it explains this apparent amplification, showing how stability and randomness combine to reduce distinguishability in the black-box setting.

## 2.2 Neural Network Kernels

A parallel line of work has developed kernel perspectives on neural network training. Neural tangent kernels (NTK) and neural network Gaussian processes (NNGP) characterize the infinite-width limit (Neal, 1996; Jacot et al., 2018; Lee et al., 2018), but at finite width the kernel at initialization drifts and fails to capture feature learning. This motivated kernel constructions that track the entire training trajectory rather than just the starting point. Path kernels and, in particular, the *loss path kernel* (LPK) integrate gradient features over time, providing a geometry that better reflects finite-width optimization (Chen et al., 2023; 2025). These ideas have been applied to study generalization and stability, highlighting that two models trained on neighboring datasets can be compared in a common reproducing kernel Hilbert space (RKHS). Our analysis builds on this kernel viewpoint: we show that DP–SGD clipping and noise induce function-level stability in the LPK geometry, and we use this as the entry point for bounding black-box distinguishability.

## 2.3 Gaussianization Tools

Quantitative Gaussian approximations have been studied via the Malliavin–Stein method (Chatterjee, 2009) and second-order Gaussian Poincaré inequalities (Nourdin & Peccati, 2012). Vidotto (2020) sharpened this approach with improved constants, and Bordino et al. (2024) applied it to obtain non-asymptotic CLTs for shallow Gaussian networks. Favaro et al. (2025) further extended the analysis to deep architectures, providing quantitative CLTs for modern overparameterized models such as Variational Autoencoders (VAEs) and diffusion samplers. We build on this line by integrating quantitative CLT bounds into $f$-DP tradeoff analyses, yielding dimension-aware privacy envelopes for black-box generative models.

## 3 Preliminaries

### 3.1 Neural Tangent Kernel and Loss Path Kernel

Neural tangent kernel (NTK) (Jacot et al., 2018) is a standard tool for analyzing wide networks. For $f_\theta : \mathbb{R}^d \to \mathbb{R}$ with parameters $\theta$, the NTK at initialization is

$$K_{\mathrm{NTK}}(x, x') = \left\langle \nabla_\theta f_\theta(x)\big|_{\theta_0}, \ \nabla_\theta f_\theta(x')\big|_{\theta_0} \right\rangle,$$

where $\theta_0$ is the random initialization. In the infinite-width limit this kernel stays constant during training, yielding linearized dynamics. In finite width, however, the NTK at initialization may not capture the full trajectory geometry. *Loss Path Kernels* (LPK) (Chen et al., 2023) address this by integrating gradient features over the entire training path:

$$K_{\ell, T}(z, z') = \int_0^T \left\langle \nabla_w \ell(w(t), z), \ \nabla_w \ell(w(t), z') \right\rangle dt,$$

where $w(t)$ is the parameter trajectory and $\ell$ the per-example loss. LPK reflects evolving geometry and gives sharper generalization and stability guarantees (Chen et al., 2023; 2025).

### 3.2 $f$-Differential Privacy

Differential privacy (DP) (Dwork, 2006) is the standard privacy notion. Dong et al. (2022) introduced the $f$-DP framework, which characterizes privacy via hypothesis testing.

**Definition 1** ($f$-DP)**.** *A mechanism $\mathcal{M}$ satisfies $f$-DP if for all neighboring $D \sim D'$,*

$$T(P, Q)(\alpha) \ \geq \ f(\alpha), \quad \forall \alpha \in [0, 1],$$

*where $P = \mathcal{M}(D), Q = \mathcal{M}(D')$, and $T(P, Q)(\alpha)$ is the minimal type-II error at type-I error $\leq \alpha$.*

Gaussian DP (GDP) is the special case where $f$ is the tradeoff curve between $\mathcal{N}(0, 1)$ and $\mathcal{N}(\mu, 1)$ with privacy parameter $\mu$. Pierquin et al. (2025) proposes a key property which is robustness to approximation errors:

**Lemma 1** (Total Variation Robustness). *If $d_{\text{TV}}(P, \tilde{P}) \leq \gamma$ and $d_{\text{TV}}(Q, \tilde{Q}) \leq \gamma$, then for all $\alpha \in (\gamma, 1 - \gamma)$,*

$$T(P, Q)(\alpha) \geq T(\tilde{P}, \tilde{Q})(\alpha + \gamma) - \gamma.$$

Thus one may approximate complex output distributions by Gaussian surrogates, losing only $O(\gamma)$ in tightness.

### 3.3 Differentially Private Stochastic Gradient Descent

DP–SGD (Abadi et al., 2016) is the default for training private deep models. At each iteration $t$, a minibatch $\mathcal{B}_t$ (Poisson rate $q = b/n$) is drawn, per-example gradients are clipped, and Gaussian noise is added:

$$g_t = \frac{1}{b} \sum_{\xi \in \mathcal{B}_t} \text{clip}_C(\nabla_w \ell(w_t; \xi)), \quad \tilde{g}_t = g_t + \sigma \frac{C}{b} \zeta_t, \; \zeta_t \sim \mathcal{N}(0, I),$$

$$w_{t+1} = w_t - \eta_t \tilde{g}_t.$$

Here $C$ is the clipping threshold, $\sigma$ the noise multiplier, and $\eta_t$ the learning rate. Privacy can be tracked through accountant methods such as RDP (Mironov, 2017), PRVs (Gopi et al., 2021) or $f$-DP. For our purposes, clipping and noise enforce training stability, which links DP–SGD with privacy amplification under latent randomness.

## 4 The Mechanisms of Privacy Amplification in Neural Networks

### 4.1 Problem Definition and Assumptions

We focus on a *single-release* black-box mechanism first. Neighboring datasets $D$ and $D'$ are trained with DP–SGD under the same initialization. At deployment, the trained model produces an output $X_D = f_{w_D}(Z)$, and $X_{D'} = f_{w_{D'}}(Z)$, where $f_w$ is the generative model, $Z \sim \mathcal{N}(0, I_d)$ is a latent Gaussian input hidden from the adversary. The adversary does not access $X$ directly, but evaluates a fixed scalar score $s(\cdot)$ (e.g. a calibration loss or a linear probe), giving the released quantities

$$Y_D = s(X_D), \qquad Y_{D'} = s(X_{D'}).$$

Although we present the scalar case for clarity, the extension to vector outputs $X \in \mathbb{R}^m$ is justified by a later Gaussianization step: once the two distributions are approximated by equal-variance Gaussians, the effective separation depends only on a one-dimensional projection $\langle u, X \rangle$. More generally, our results apply to any fixed scalar score $s(X)$ with bounded output derivative (including linear probes), and the ensuing RKHS bounds are uniform in $u$. Thus the scalar analysis provides valid bounds for vector outputs while keeping the notation simple. We adopt the standard coupled execution (identical mini-batches and DP noise for $D$ and $D'$), which affects only the analysis and not the mechanism itself. More detailed discussion can be found at Appendix B.2.

The LPK aggregates gradient features along the optimization path,

$$K_{\ell,T}(z, z') = \int_0^T \langle \nabla_w \ell(w(t), z), \nabla_w \ell(w(t), z') \rangle dt,$$

and provides a natural geometry for comparing outputs induced by different datasets. Since the LPK depends on the dataset, we place both $D$ and $D'$ in the common RKHS of the dominating kernel

$$K_{\ell,T}^{\oplus} := K_{\ell,T}^D + K_{\ell,T}^{D'}.$$

All RKHS norms and kernel expectations in what follows are taken with respect to $K_{\ell,T}^{\oplus}$.

**Assumption 1** (Model and Gaussian regularity). *$f_\theta$ is a finite-width, differentiable neural network with 1-Lipschitz activations. The release score $s : \mathbb{R}^{\text{out}} \to \mathbb{R}$ is $\beta$-smooth in its argument and has bounded gradient $\|\nabla_x s(x)\| \leq \Lambda_s$. For $Z \sim \mathcal{N}(0, I_d)$, let $Y_T(Z) := s(f_{w_T}(Z))$. We assume the following moments are finite:*

$$\mathbb{E}\big\|\nabla_z Y_T(Z)\big\|^4 < \infty, \qquad \mathbb{E}\big\|\nabla_z^2 Y_T(Z)\big\|_{\text{HS}}^4 < \infty,$$

*where gradients in $z$ act on the composite map $z \mapsto s(f_{w_T}(z))$.*

**Assumption 2** (Near-isotropic kernel scale in effective dimension). *There exists $C_T > 0$ such that for probes $W, W' \overset{\text{i.i.d.}}{\sim} \Pi$ (the sampling law used at release time), the following scaling relations hold for the loss path kernel built from the training loss $\ell_{train}$:*

$$\mathbb{E}K_{\ell,T}^{\oplus}(W, W) = \Theta(C_T \, d_{\text{eff}}), \qquad \mathbb{E}_{W,W'} K_{\ell,T}^{\oplus}(W, W') = \Theta(C_T),$$

*where $d_{\text{eff}} := \mathbb{E}K_{\ell,T}^{\oplus}(W, W) \big/ \mathbb{E}_{W,W'} K_{\ell,T}^{\oplus}(W, W')$.*

**Assumption 3** (Lipschitz continuity of loss gradients). *There exists $L_\ell < \infty$ such that for all $t \in [0, T]$ and inputs $z$,*

$$\|\nabla_w \ell_{train}(w_D(t); z) - \nabla_w \ell_{train}(w_{D'}(t); z)\| \leq L_\ell \|w_D(t) - w_{D'}(t)\|.$$

*This holds with $L_\ell \leq \Lambda_{train} L_g$ if $\|\nabla_f \ell_{train}(f, z)\| \leq \Lambda_{train}$ and $\|\nabla_w f(w, z) - \nabla_w f(w', z)\| \leq L_g \|w - w'\|$.*

Assumption 1 provides the Gaussian regularity for the released scalar $Y_T(Z) = s(f_{w_T}(Z))$, which we use in the Gaussianization step and in controlling the concentration of the relevant quantities. Assumption 2 relates the LPK's diagonal scale to the effective input dimension $d_{\text{eff}}$ and ensures that the variance side does not degenerate. Assumption 3 controls the Lipschitz continuity of per-example *training-loss* gradients along the path, which is used in the stability recursion.

## 4.2 PATH KERNEL STABILITY IN DP–SGD

To connect path-wise stability of training losses with black-box scores, we pass from the loss path kernel to score path kernels. By the chain rule $\nabla_w s(w; z) = (\partial_f s)(f_w(z), z) \nabla_w f(w, z)$, and if $\|\partial_f s\| \leq \Lambda_s$ while the network satisfies Assumption 1, then the score kernel

$$K_{s,T}(z, z') := \int_0^T \langle \nabla_w s(w(t); z), \, \nabla_w s(w(t); z') \rangle dt$$

is dominated in Loewner order by the loss kernel: $K_{s,T} \preceq c_s^2 K_{\ell,T}$ for some $c_s = O(\Lambda_s)$. Hence

$$\|\Delta s\|_{\mathcal{H}(K_{s,T}^{\oplus})} \leq c_s \|\Delta \ell\|_{\mathcal{H}(K_{\ell,T}^{\oplus})},$$

so any LPK-space stability bound automatically upper-bounds discrepancies of admissible scores.

This reduction motivates carrying out the analysis directly in $\mathcal{H}(K_{\ell,T}^{\oplus})$: under clipping, subsampling, and Gaussian noise, neighboring datasets yield generators whose score discrepancies are controlled at the function level, without requiring their parameter trajectories to remain close.

In DP–SGD, each step uses Poisson sampling with rate $q = b/n$, applies clipping with threshold $C > 0$ to per-example gradients, and adds Gaussian noise at the mini-batch level. In what follows we focus on the fixed subsampling rate. For neighboring datasets $D, D'$, we use a coupled execution with identical mini-batches and noise.

Let $E_T^D := \int_0^T \|\dot{w}_D(t)\|^2 \, dt$ and $E_T^{D'} := \int_0^T \|\dot{w}_{D'}(t)\|^2 \, dt$, and write $E_T^{\oplus} := E_T^D + E_T^{D'}$. Set $\|\eta\|_2 = (\sum_{t=0}^{T-1} \eta_t^2)^{1/2}$, where $\{\eta_t\}$ is the learning-rate schedule.

**Proposition 1** (LPK stability under DP–SGD). *Under Assumption 3 and the coupled execution,*

$$\mathbb{E}\|\Delta \ell\|_{\mathcal{H}(K_{\ell,T}^{\oplus})} \leq \frac{B_T}{n} \tag{1}$$

*with*

$$B_T = 2C \left( \sqrt{E_T^{\oplus}} \, \|\eta\|_2 \, e^{\, c_1 L_\ell^2 \|\eta\|_2^2} + \sqrt{\frac{\sum_t \eta_t^2}{q}} \right), \tag{2}$$

*where $c_1 > 0$ is a constant independent of $n, T$. If $\sum_t \eta_t^2 = O(1)$, then $e^{\, c_1 L_\ell^2 \|\eta\|_2^2} = O(1)$.*

*Proof Sketch.* Write $\phi_t^D(z) = \nabla_w \ell(w_D(t); z)$ and similarly for $D'$. Along each path,

$$\ell(w_T; z) - \ell(w_0; z) = \int_0^T \langle \phi_t^D(z), \dot{w}_D(t) \rangle dt,$$

and the difference between $D$ and $D'$ admits the symmetric representation in the RKHS of $K_{\ell,T}^\oplus$. Cauchy–Schwarz in feature space gives $\|\Delta\ell\|_{\mathcal{H}(K_{\ell,T}^\oplus)} \leq \sqrt{E_T^\oplus}$ times a factor controlled by the velocity discrepancy. The one-step recursion for $\Delta_t = w_t(D) - w_t(D')$,

$$\mathbb{E}\|\Delta_{t+1}\|^2 \leq (1 + cL^2\eta_t^2)\mathbb{E}\|\Delta_t\|^2 + c\,\eta_t^2\,(2C/b)^2 q,$$

avoids any Grönwall factor in $S_T$; iterating yields $\mathbb{E}\sum_t \eta_t^2\|g_t - g_t'\|^2 \lesssim \left(\frac{2C}{n}\right)^2 e^{O(L^2\|\eta\|_2^2)} \sum_t \eta_t^2 + \left(\frac{2C}{n}\right)^2 \frac{\sum_t \eta_t^2}{q}$. Combining with the feature-space bound gives equation 1. $\qquad\square$

Proposition 1 shows that, under clipping, noise, and subsampling, the LPK distance between models trained on neighboring datasets decays at rate $1/n$. The dependence on the learning-rate schedule enters through $\sum_t \eta_t^2$, which remains $O(1)$ under standard decaying schedules. This function-level stability is the interface to the high-dimensional Gaussian input geometry used in the next section.

### 4.3 Gaussian Approximation Error

To connect stability bounds with testing tradeoffs, we approximate the law of each scalar release by a Gaussian with matched mean and variance, and quantify the resulting approximation error.

Let $Z \sim \mathcal{N}(0, I_d)$ be independent of the training data and consider a scalar release

$$Y_D = s\big(f_{w_D}(Z)\big) \in \mathbb{R}, \qquad Y_{D'} = s\big(f_{w_{D'}}(Z)\big) \in \mathbb{R},$$

where $f_{w_T}$ is the trained generator and $s$ is the fixed scalar score used at release time. Write $\mu_D = \mathbb{E}Y_D$, $\sigma_D^2 = \mathrm{Var}(Y_D)$ (and analogously for $D'$). Define the Gaussian approximation error

$$\gamma_d := \max\left\{d_{\mathrm{TV}}\big((Y_D - \mu_D)/\sigma_D,\ \mathcal{N}(0,1)\big),\ d_{\mathrm{TV}}\big((Y_{D'} - \mu_{D'})/\sigma_{D'},\ \mathcal{N}(0,1)\big)\right\}.$$

We record two complementary total-variation bounds for the single-release scalar output $Y$. The first applies to deterministic maps $Y = g(Z)$ and rests on second-order Poincaré (Nourdin & Peccati, 2012) on Gaussian space (in our setting one may take $g(z) = s(f_{w_T}(z))$). The second exploits a *conditionally Gaussian* (variance–mixture) structure that captures many samplers (VAE decoders, diffusion reverse steps, temperature noise), and yields an $O(1/d_{\mathrm{eff}})$ rate under mild variance–concentration. While the variance–mixture bound aligns naturally with modern generators and is easy to verify, the second-order Poincaré bound can deliver faster asymptotic rates when higher-order derivative moments are controlled; the two are thus complementary.

**Proposition 2** (Generic bound via second-order Poincaré). *Let $Z \sim \mathcal{N}(0, I_d)$ and $Y = g(Z)$ with $\sigma^2 := \mathrm{Var}(Y) > 0$. Define*

$$L_x := \Big(\mathbb{E}\|\nabla_z g(Z)\|^4\Big)^{1/4}, \qquad H_x := \Big(\mathbb{E}\|\nabla_z^2 g(Z)\|_{\mathrm{HS}}^4\Big)^{1/4}.$$

*Then*

$$d_{\mathrm{TV}}\big(\mathcal{L}(Y), \mathcal{N}(\mathbb{E}Y, \sigma^2)\big) = O\Big(\frac{L_x H_x}{\sigma^2}\Big).$$

*In particular, if $L_x = O(1)$ and $H_x = O(d_{\mathrm{eff}}^{-\beta})$ for some $\beta \in (0,1]$ and $\sigma^2 = \Theta(d_{\mathrm{eff}})$, then $d_{\mathrm{TV}} = O\big(d_{\mathrm{eff}}^{-(1+\beta)}\big)$.*

Inspired by Favaro et al. (2025), a more specific and less assumption-intensive bound for conditionally Gaussian outputs is as follows.

**Proposition 3** (Variance–mixture bound for conditionally Gaussian outputs). *Suppose that, for some auxiliary sampler state $S$, the release is conditionally Gaussian with $A = \sigma^2(S) \geq 0$:*

$$Y \mid S \sim \mathcal{N}\big(0, A\big), \qquad \sigma^2 := \mathrm{Var}(Y) = \mathbb{E}[A] > 0.$$

*Then*

$$d_{\mathrm{TV}}\big(\mathcal{L}(Y), \mathcal{N}(0, \sigma^2)\big) \leq \frac{8\,\mathrm{Var}(A)}{\sigma^4} = O\Big(\frac{\mathrm{Var}(A)}{\sigma^4}\Big).$$

*In particular, if $\mathrm{Var}(Y) = \Theta(d_{\mathrm{eff}})$ and $\mathrm{Var}(A) = O(d_{\mathrm{eff}})$, then $d_{\mathrm{TV}} = O(d_{\mathrm{eff}}^{-1})$.*

**Remark.** *Proposition 2 is assumption-light in structure but depends on curvature moments of g; when specified assumptions are hold, it is potentially faster than $1/d_{\text{eff}}$. Proposition 3 relies on a conditional-Gaussian structure; this structure holds in many generators (e.g., VAEs, diffusion with Gaussian injections), making the bound practically convenient though sometimes looser asymptotically. Both bounds are scale-invariant through the $\sigma^{-2}$ factors and integrate seamlessly with our $f$-DP robustness transfer.*

We next combine Gaussianization with an equal-variance alignment. For a scalar $Y$ with $\text{Var}(Y) = \sigma^2 > 0$, define the standardized Gaussianization error

$$\gamma(Y) := d_{\text{TV}}\big((Y - \mathbb{E}Y)/\sigma, \, \mathcal{N}(0,1)\big).$$

Given the releases $Y_D = s(f_{w_D(T)}(Z))$ and $Y_{D'} = s(f_{w_{D'}(T)}(Z))$, set

$$\gamma_d := \max\big\{\gamma(Y_D), \, \gamma(Y_{D'})\big\}.$$

To compare with equal-variance Gaussian surrogates, write

$$\sigma_D^2 := \text{Var}(Y_D), \quad \sigma_{D'}^2 := \text{Var}(Y_{D'}), \quad \bar\sigma^2 := \tfrac{1}{2}(\sigma_D^2 + \sigma_{D'}^2).$$

Define the equal-variance slack

$$\omega := c_0 \, \frac{|\sigma_D^2 - \sigma_{D'}^2|}{\bar\sigma^2}, \tag{3}$$

for a universal constant $c_0 > 0$. Its scaling can be bounded in expectation as $\omega = \tilde{O}\big(1/(n\sqrt{C_T d_{\text{eff}}})\big)$, as established in Lemma 8.

**Lemma 2** (Reduction to equal-variance Gaussian surrogates). *With $\gamma_d$ and $\omega$ as above, for any $\alpha \in (\gamma_d + \omega, \, 1 - \gamma_d - \omega)$,*

$$T\big(P_{Y_D}, P_{Y_{D'}}\big)(\alpha) \geq T\big(\mathcal{N}(\mathbb{E}Y_D, \bar\sigma^2), \mathcal{N}(\mathbb{E}Y_{D'}, \bar\sigma^2)\big)\big(\alpha + \gamma_d + \omega\big) \, - \, (\gamma_d + \omega).$$

*Proof Sketch.* Apply TV–robustness to pass from $(P_{Y_D}, P_{Y_{D'}})$ to $(\mathcal{N}(\mathbb{E}Y_D, \sigma_D^2), \mathcal{N}(\mathbb{E}Y_{D'}, \sigma_{D'}^2))$ incurring a horizontal/vertical shift $\gamma_d$; then replace each marginal by its equal-variance version, which adds $\omega$. $\qquad\square$

### 4.4 FROM PATH STABILITY TO TEST INDISTINGUISHABILITY

We now switch from path-wise stability to the indistinguishability of the released scores $Y_D = s(f_{w_D(T)}(W))$ and $Y_{D'} = s(f_{w_{D'}(T)}(W))$ under probe inputs $W \sim \Pi$ (as in Assumption 2). The key quantity is the GDP parameter that controls the optimal tradeoff curve. Throughout this subsection the RKHS is the common space $\mathcal{H}\big(K_{\ell,T}^{\oplus}\big)$ introduced in §4.1, and kernel expectations are taken with respect to $K_{\ell,T}^{\oplus}$.

**Theorem 1** (Privacy amplification under effective dimension). *Under Assumptions 1–2, with fixed subsampling rate $q = b/n$, and with $\gamma_d, \omega$ defined above, for any $\alpha \in (\gamma_d + \omega, \, 1 - \gamma_d - \omega)$,*

$$T\big(P_{Y_D}, P_{Y_{D'}}\big)(\alpha) \geq G_{\mu_{\text{eff}}}(\alpha + \gamma_d + \omega) \, - \, (\gamma_d + \omega), \qquad \mu_{\text{eff}} \lesssim \frac{\|\Delta f\|_{\mathcal{H}(K_{\ell,T}^{\oplus})}}{\sqrt{\mathbb{E}\, K_{\ell,T}^{\oplus}(W,W)}}.$$

*Combining Proposition 1 with Assumption 2 yields the upper-envelope scaling*

$$\mathbb{E}\, \mu_{\text{eff}} \lesssim \frac{1}{n\sqrt{C_T \, d_{\text{eff}}}}.$$

*Proof Sketch.* By Lemma 2, it suffices to compare the equal-variance Gaussian pair $\mathcal{N}(\mathbb{E}Y_D, \bar\sigma^2)$ vs. $\mathcal{N}(\mathbb{E}Y_{D'}, \bar\sigma^2)$ at the shifted level $\alpha + \gamma_d + \omega$, whose tradeoff is $G_\mu$ with $\mu = |\mathbb{E}Y_D - \mathbb{E}Y_{D'}|/\bar\sigma$. Bounding the mean gap and the variance scale in $\mathcal{H}(K_{\ell,T}^{\oplus})$ via Lemmas 9–10 and Assumption 2 gives the displayed $\mu_{\text{eff}}$. Injecting Proposition 1 finishes the proof. $\qquad\square$

The quantities $\gamma_d$ and $\omega$ capture the Gaussianization error and the equal-variance adjustment, which act only as small horizontal/vertical shifts in the tradeoff curve (Lemma 2). Typically $\gamma_d = O(1/d_{\text{eff}})$ (or faster under smoothness) and $\mathbb{E}[\omega] = \tilde{O}(1/(n\sqrt{C_T d_{\text{eff}}}))$, so both are higher-order. The dominant term is the effective parameter: while Theorem 1 certifies $\mu_{\text{eff}} = O(d_{\text{eff}}^{-1/2})$, in

practice it often decays faster, since $C_T$ may grow with $d_{\text{eff}}$, near-isotropy may hold with slack, and DP noise further reduces $\|\Delta f\|_{\mathcal{H}}$. Thus empirical curves usually lie below the $d_{\text{eff}}^{-1/2}$ guide, and the theorem should be read as an upper envelope rather than a tight rate.

### 4.5 COMPOSITION ACROSS MULTIPLE RELEASES

We now quantify black-box distinguishability when $m$ i.i.d. probe inputs are released. Let $W_1, \ldots, W_m \overset{\text{i.i.d.}}{\sim} \Pi$ and define

$$\mathbf{Y}_D = \big(s(f_{w_D}(W_1)), \ldots, s(f_{w_D}(W_m))\big), \mathbf{Y}_{D'} = \big(s(f_{w_{D'}}(W_1)), \ldots, s(f_{w_{D'}}(W_m))\big).$$

From Theorem 1, we have the single-release envelope with Gaussianization slack $\gamma_d$ and equal-variance slack $\omega$ from §4.3. Write $\varepsilon := \gamma_d + \omega$. For $m$ releases, the product-level slack

$$\Gamma_m := \max\Big\{ d_{\text{TV}}\big(P_{\mathbf{Y}_D}, \tilde{P}^{\otimes m}\big), d_{\text{TV}}\big(P_{\mathbf{Y}_{D'}}, \tilde{Q}^{\otimes m}\big)\Big\}$$

$$+ \max\Big\{ d_{\text{TV}}\big(\tilde{P}^{\otimes m}, \bar{P}^{\otimes m}\big), d_{\text{TV}}\big(\tilde{Q}^{\otimes m}, \bar{Q}^{\otimes m}\big)\Big\}, \tag{4}$$

compares the true distributions to their Gaussian and equal-variance surrogates. By a hybrid argument and Hellinger tensorization (App. C.4), there exists a constant $c_1 > 0$ such that

$$\Gamma_m \leq c_1 \min\big\{ m\varepsilon, \ 2\sqrt{m\varepsilon} \big\}. \tag{5}$$

**Theorem 2** (GDP envelope for $m$ releases with explicit slacks)**.** *Under the assumptions of Theorem 1 and independent probe draws across releases, for any $\alpha \in (\Gamma_m, 1 - \Gamma_m)$,*

$$T\big(P_{\mathbf{Y}_D}, P_{\mathbf{Y}_{D'}}\big)(\alpha) \geq G_{\mu_{\text{eff}}^{(m)}}\big(\alpha + \Gamma_m\big) - \Gamma_m, \qquad \mu_{\text{eff}}^{(m)} \leq \sqrt{m}\,\mu_{\text{eff}}^{(1)}. \tag{6}$$

*Moreover, combining Theorem 1 with Assumption 2,*

$$\mathbb{E}\,\mu_{\text{eff}}^{(m)} \lesssim \frac{\sqrt{m}}{n\,\sqrt{C_T\,d_{\text{eff}}}}.$$

From Proposition 3 or Proposition 2, one typically has $\gamma_d = O(1/d_{\text{eff}})$ or $O\big(d_{\text{eff}}^{-(1+\beta)}\big)$, while (cf. App. C.3) $\omega = \tilde{O}\big(1/(n\sqrt{C_T\,d_{\text{eff}}})\big)$ in expectation. Hence, for large $d_{\text{eff}}$ and moderate $m$,

$$\Gamma_m = O\Big( \min\{ m/d_{\text{eff}}, \ \sqrt{m/d_{\text{eff}}} \}\Big), \tag{7}$$

with the dominant contribution from $\gamma_d$. Together, these bounds show that the signal side composes sublinearly ($\mu_{\text{eff}}^{(m)} \propto \sqrt{m}$), while the slack side remains small as long as $m \ll d_{\text{eff}}$, so that the overall envelope remains close to the ideal GDP curve.

## 5 SIMULATION AND EMPIRICAL EVIDENCE

We provide a minimal set of simulations to illustrate the behavior predicted by our theory. All experiments are based on a small VAE trained on MNIST using DP-SGD. Quantities in Theorem 1 are estimated post hoc using Gaussian probes $Z \sim \mathcal{N}(0, I_d)$ and a smooth calibration score, without per-example access. Implementation details are deferred to Appendix D.1.

**Envelope vs. empirical tradeoff.** Figure 2a compares our adjustable GDP envelope with the formal accountant baseline and the empirical black-box MIA result by Annamalai et al. (2024). The new bound lies above the baseline and matches the empirical curve more closely, showing that LPK-based stability combined with latent randomness captures the observed amplification effect.

**Scaling with effective dimension and dataset size.** Our theory predicts two complementary decay behaviors. First, Theorem 1 implies that $\mu_{\text{eff}}$ decreases at least as $O(d_{\text{eff}}^{-1/2})$ under near-isotropy. Figure 2b confirms this trend: $\hat{\mu}_{\text{eff}}$ decreases monotonically with $d_{\text{eff}}$, and the empirical slope is often steeper, consistent with the remark following Theorem 1. Second, Proposition 1 predicts that the LPK discrepancy decays as $1/n$. Figure 2c supports this prediction: Monte Carlo estimates of $\|\Delta f\|_{\mathcal{H}}$ follow an approximate $1/n$ slope on log–log axes, with mild deviations due to the learning-rate schedule.

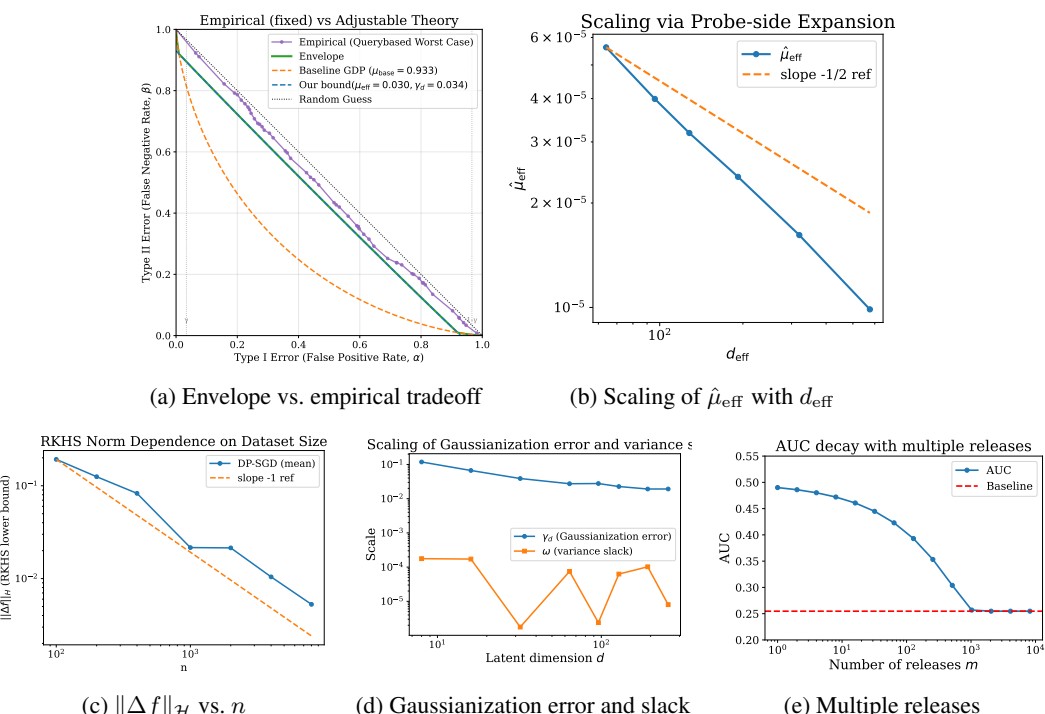

(a) Envelope vs. empirical tradeoff      (b) Scaling of $\hat{\mu}_{\text{eff}}$ with $d_{\text{eff}}$

(c) $\|\Delta f\|_{\mathcal{H}}$ vs. $n$    (d) Gaussianization error and slack    (e) Multiple releases

Figure 2: Simulation results validating the theoretical analysis.

**Gaussianization and variance slacks.** Propositions 2 and 3 yield bounds on the Gaussian approximation error $\gamma_d$, while Lemma 2 introduces the equal-variance slack $\omega$. Figure 2d shows both quantities remain small in practice ($\gamma_d = O(1/d_{\text{eff}})$ or faster; $\omega$ further suppressed by $1/n$), supporting their role as higher-order correction terms in the GDP reduction.

**Multiple releases.** Theorem 2 states that the effective parameter composes sublinearly, $\mu_{\text{eff}}^{(m)} \leq \sqrt{m}\,\mu_{\text{eff}}^{(1)}$, with product-level slack $\Gamma_m$ small whenever $m \ll d_{\text{eff}}$. We illustrate this relationship in Figure 2e. We use the Area Under the Curve (AUC) of the tradeoff curve to measure privacy protection, which is $A(T) := \int_0^1 T(\alpha)\,d\alpha$. A larger value indicates stronger privacy protection. As $m$ increases, privacy protection gradually weakens, converging to the baseline level at a certain threshold. This is primarily because the increase in $\Gamma_d$ causes the new tradeoff curve to gradually drift overall until it no longer contributes to the envelope.

## 6 CONCLUSION

We revisited privacy amplification in differentially private generative models from a test-centric $f$-DP perspective. By combining function-level stability under DP–SGD with high-dimensional latent randomness, we derived GDP-style envelopes that capture the empirically observed gap between black-box MIAs and worst-case accounting. Our results highlight that distinguishability decays with dataset size and effective input dimension, and that stability can be tracked post hoc using path kernels and Gaussian probes. Together, these findings offer a principled explanation for why DP-trained generators often appear more private in practice, and provide a quantitative tool for guiding model selection, hyperparameter tuning, and risk assessment in realistic pipelines.

A key direction for future research is the development of sharper Gaussian approximation techniques and more refined treatments of composition effects. This is especially critical for conditional generative models, where auxiliary variables could amplify attack power. Such research holds the promise of yielding both tighter theoretical guarantees and broader practical applicability.

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

## A  GENERAL STATEMENTS AND BROADER CONTEXT

### A.1  REPRODUCIBILITY STATEMENT

To ensure the reproducibility of our work, we have included all necessary materials in the appendices and supplementary files. Complete proofs for all theoretical results are available in the appendix. The supplementary material contains the full source code for our numerical simulations and experiments. Further details on the experimental setup and hyperparameters are provided in Appendix D.1.

### A.2  DECLARATION ON THE USE OF AI-ASSISTED TECHNOLOGIES

During the preparation of this work, the authors used ChatGPT and Gemini as writing assistants to enhance the text's readability and linguistic accuracy. The use of this tool was confined to improving grammar and rephrasing sentences for clarity. All AI-generated suggestions were reviewed and edited by the authors, who take full responsibility for the scientific integrity and entire content of this publication, as stipulated by the ICLR Code of Ethics.

## B  THEORETICAL BACKGROUND AND SETUP

### B.1  A PRIMER ON LOSS PATH KERNELS (LPK)

One of the core analytical tools in this paper is the Loss Path Kernel (LPK) (Chen et al., 2023). We provide a brief background and clarifies its role in our work.

#### B.1.1  MOTIVATION: BEYOND STATIC KERNELS

A well-known tool in the theoretical analysis of neural networks is the Neural Tangent Kernel (NTK) (Jacot et al., 2018). The NTK characterizes the evolution of infinite-width neural networks under gradient descent, revealing a deep connection to kernel methods. However, a primary limitation of NTK theory is its reliance on the infinite-width limit, where network parameters are assumed to stay close to their initialization values. This makes the NTK a **static kernel**, largely fixed at the beginning of training, which struggles to fully capture the complex feature learning that occurs in finite-width networks during practical training.

The LPK was introduced to overcome this limitation. Unlike the NTK, the LPK is a dynamic, data- and algorithm-dependent kernel. It is not fixed at a single point in time but is defined by integrating gradient information over the entire optimization path. This allows the LPK to capture the full training dynamics, offering a more precise and informative lens for analyzing the generalization and stability of neural networks.

#### B.1.2  DEFINITION OF THE LPK

The central idea of the LPK is to measure the similarity between two data points by comparing their respective loss gradients throughout the entire training process.

**Definition 2** (Loss Path Kernel (LPK)). *Suppose the parameters of a neural network, $\boldsymbol{w}$, follow a continuous path $\boldsymbol{w}(t)$ for $t \in [0, T]$, determined by a training set $\mathcal{S}$ and an optimization algorithm. For any two data points $\boldsymbol{z}$ and $\boldsymbol{z}'$, the Loss Path Kernel along this path is defined as:*

$$K_T(\boldsymbol{z}, \boldsymbol{z}'; \mathcal{S}) \triangleq \int_0^T \langle \nabla_{\boldsymbol{w}} l(\boldsymbol{w}(t), \boldsymbol{z}), \nabla_{\boldsymbol{w}} l(\boldsymbol{w}(t), \boldsymbol{z}') \rangle dt,$$

*where $l(\boldsymbol{w}(t), \boldsymbol{z})$ is the loss of the model with parameters $\boldsymbol{w}(t)$ on data point $\boldsymbol{z}$, and $\langle \cdot, \cdot \rangle$ denotes the inner product of the gradient vectors.*

By definition, the LPK is a valid kernel function as it is symmetric and its corresponding kernel matrix is positive semi-definite.

### B.1.3 CONNECTION TO RKHS

As a valid kernel, the LPK naturally defines a RKHS. More importantly, it establishes a key equivalence: the loss function of a neural network trained by gradient flow at any time $T$ is equivalent to a general kernel machine equipped with the LPK. This equivalence is powerful because it allows the analysis of complex neural network dynamics to be translated into the analysis of functions within the more structured framework of an RKHS.

### B.1.4 ROLE IN THIS WORK

In our study of differential privacy, the LPK plays an indispensable role. A central thesis of this paper is that the DP-SGD algorithm induces **function-level stability**. This means that when a model is trained on two neighboring datasets, $\mathcal{D}$ and $\mathcal{D}'$, the resulting functions $f_{\boldsymbol{w}_\mathcal{D}}$ and $f_{\boldsymbol{w}_{\mathcal{D}'}}$ are close in a function space, even if their parameters $\boldsymbol{w}_\mathcal{D}$ and $\boldsymbol{w}_{\mathcal{D}'}$ are far apart in parameter space.

The LPK provides the theoretical framework to precisely quantify this notion of "function-level closeness." Specifically:

- **Provides the Geometric Space:** The RKHS defined by the LPK is the appropriate geometry for measuring the distance between the two model functions. Our analysis is grounded in the common RKHS of the dominating kernel $K_{\ell,T}^\oplus$.

- **Connects Training Dynamics to Output Stability:** Our key stability result, Proposition 1, is derived directly from analyzing the DP-SGD dynamics within the LPK RKHS. By bounding the norm distance $\|\Delta \ell\|_{\mathcal{H}(K_{\ell,T}^\oplus)}$ between the two loss functions in this space, we can directly derive bounds on the distinguishability of the models' outputs.

- **Frees the Analysis from Parameter Proximity:** By using the LPK, our analysis does not need to assume that $\|\boldsymbol{w}_\mathcal{D} - \boldsymbol{w}_{\mathcal{D}'}\|$ is small. This makes our framework better suited to the non-convex optimization landscape of deep learning and provides a more solid foundation for explaining the privacy amplification effects observed in black-box settings.

In summary, the LPK serves as the crucial bridge connecting the dynamics of DP-SGD training to the black-box privacy guarantees of the final generative model.

### B.2 ADDITIONAL DETAILS ABOUT THE PROBLEM DEFINITION AND ASSUMPTIONS

In the main text we distinguished between the raw model output

$$X_D = f_{w_D}(Z), \qquad X_{D'} = f_{w_{D'}}(Z),$$

and the released scalar values

$$Y_D = s(X_D), \qquad Y_{D'} = s(X_{D'}).$$

Here $f_w$ is the generative model, $Z \sim \mathcal{N}(0, I_d)$ is the latent input, and $s(\cdot)$ is a fixed score used by the adversary. This separation emphasizes that the actual black-box outputs can be high-dimensional (e.g. images), while our analysis concerns scalar statistics derived from them.

Although we focus on the scalar case in the main text, this is without loss of generality. For vector outputs $Y \in \mathbb{R}^m$, after Gaussianization, the problem reduces to equal-variance Gaussians, for which

the optimal test is linear, and the separation parameter is

$$\mu_{\text{vec}} = \sup_{u \in \mathbb{S}^{m-1}} \frac{|\mathbb{E}\langle u, Y_D\rangle - \mathbb{E}\langle u, Y_{D'}\rangle|}{\sqrt{\text{Var}(\langle u, Y\rangle)}}.$$

Since our RKHS bounds hold uniformly over $u$, the scalar analysis extends directly to vectors.

The loss path kernel (LPK) used in the analysis is defined in terms of the per-example training loss $\ell_{\text{train}}$, not the release score $s$. It aggregates per-example gradients along the optimization path,

$$K_{\ell,T}(z, z') = \int_0^T \langle \nabla_w \ell_{\text{train}}(w(t); z), \nabla_w \ell_{\text{train}}(w(t); z')\rangle dt,$$

and provides a geometry for measuring the discrepancy between neighboring training runs. The bridge to $s$ is given by bounded output derivatives: if $\|\nabla_f s\| \leq \Lambda$, then differences in $Y_D, Y_{D'}$ can be controlled (up to $\Lambda$) by the same $\mathcal{H}(K_{\ell,T}^\oplus)$ norm that governs stability under $\ell_{\text{train}}$. This explains why the subsequent analysis, although stated in terms of the LPK, applies to the black-box releases $Y_D, Y_{D'}$.

In the problem definition we fixed $Z \sim \mathcal{N}(0, I_d)$ for clarity. Assumption 2 is stated more generally for probes $W, W' \sim \Pi$, where $\Pi$ is the distribution used at release time. This formulation covers Gaussian inputs as well as samplers with injected noise or variance mixtures. The ratio

$$d_{\text{eff}} := \frac{\mathbb{E} K_{\ell,T}^\oplus(W, W)}{\mathbb{E}_{W,W'} K_{\ell,T}^\oplus(W, W')}$$

plays the role of an effective input dimension, ensuring that the variance of released outputs does not degenerate as dimension grows.

Finally, we recall the roles of the assumptions. Assumption 1 provides Gaussian regularity for Malliavin–Stein bounds and for concentration of the LPK scale. Assumption 2 ties the kernel's diagonal scale to $d_{\text{eff}}$, guaranteeing non-trivial output variance. Assumption 3 controls Lipschitz continuity of loss gradients along training, which is used to bound discrepancies in the stability recursion. Together, these assumptions ensure that the LPK geometry faithfully captures differences between neighboring datasets under DP–SGD training.

## C  PROOFS OF MAIN RESULTS

### C.1  PROOF OF PROPOSITION 1

We keep the setting of §4.2. For DP–SGD with Poisson subsampling rate $q = b/n$, clipping threshold $C > 0$, and batch noise multiplier $\sigma$,

$$w_{t+1} = w_t - \eta_t \left(\frac{1}{b} \sum_{\xi \in \mathcal{B}_t} \text{clip}_C(\nabla \ell(w_t; \xi)) + \sigma \frac{C}{b} \zeta_t\right), \qquad \zeta_t \sim \mathcal{N}(0, I).$$

Under the coupled execution (same mini-batches and same $\zeta_t$ on $D$ and $D'$), write

$$\Delta_t := w_t(D) - w_t(D'), \qquad \delta g_t := g_t - g_t', \qquad \delta v_t := \eta_t \delta g_t,$$

so that $\Delta_{t+1} = \Delta_t - \delta v_t$. Throughout, $c, c_1, c_2 > 0$ denote absolute constants that may change line to line.

**Discrete-time interpolation and feature maps.** On each interval $[t, t+1)$ use the piecewise-linear interpolation $w_D(s) = w_t(D) + (s - t)(w_{t+1}(D) - w_t(D))$ and similarly for $D'$; the path energies are

$$E_T^D = \int_0^T \|\dot{w}_D(s)\|^2 \, ds = \sum_{t=0}^{T-1} \|w_{t+1}(D) - w_t(D)\|^2, \qquad E_T^{D'} \text{ analogously}, \quad E_T^\oplus := E_T^D + E_T^{D'}.$$

Let $\phi_t^D(z) := \nabla_w \ell(w_D(t), z)$ and $\phi_t^{D'}(z) := \nabla_w \ell(w_{D'}(t), z)$. By definition of the LPK, the RKHS of the dominating kernel $K_{\ell,T}^\oplus := K_{\ell,T}^D + K_{\ell,T}^{D'}$ is the Hilbert direct sum $\mathcal{H}(K_{\ell,T}^D) \oplus \mathcal{H}(K_{\ell,T}^{D'})$. For

any square-integrable weight $v : [0, T] \to \mathbb{R}^{\dim(w)}$,

$$g_v^D(z) := \int_0^T \langle \phi_s^D(z), v(s) \rangle ds \ \in \ \mathcal{H}(K_{\ell,T}^D), \quad \|g_v^D\|_{\mathcal{H}(K_{\ell,T}^D)} \ \le \ \Big( \int_0^T \|v(s)\|^2 ds \Big)^{1/2}, \qquad (8)$$

and similarly for $D'$ (reproducing property plus the definition of $K_{\ell,T}$).

**Lemma 3** (One-step bound on the clipped batch means). *Under Assumption 3, for all t,*

$$\|\delta g_t\| \ \le \ L_\ell \|\Delta_t\| \ + \ \frac{2C}{b} \mathbf{1}\{\mathrm{hit}(t)\},$$

*where $\mathbf{1}\{\mathrm{hit}(t)\}$ indicates that the differing example is drawn into the batch at step t; for Poisson subsampling, $\mathbb{E}[\mathbf{1}\{\mathrm{hit}(t)\}] = q$ and $\mathbb{E}[\mathbf{1}\{\mathrm{hit}(t)\}^2] = q$.*

*Proof.* If the differing sample is not drawn, the two batches coincide and $w \mapsto \frac{1}{b} \sum_{\xi \in \mathcal{B}_t} \mathrm{clip}_C(\nabla \ell(w; \xi))$ is $L$-Lipschitz since clipping is a projection to the $C$-ball and $\nabla \ell(\cdot; \xi)$ is $L$-Lipschitz by Assumption 3. If the batches differ by one example, their means differ by at most $2C/b$. $\qquad \square$

**Lemma 4** (Quadratic-moment recursion and its solution). *Let $A_t := \mathbb{E}\|\Delta_t\|^2$. Then*

$$A_{t+1} \ \le \ \big(1 + c_1 L_\ell^2 \eta_t^2\big) A_t \ + \ c_2 \eta_t^2 (2C/b)^2 q. \qquad (9)$$

*Consequently, with $\|\eta\|_2^2 = \sum_{t=0}^{T-1} \eta_t^2$,*

$$\max_{0 \le t \le T} A_t \ \le \ c_3 \left( \frac{2C}{n} \right)^2 \exp\big(c_1 L_\ell^2 \|\eta\|_2^2\big) \sum_{s=0}^{T-1} \eta_s^2, \qquad (10)$$

$$\sum_{t=0}^{T-1} A_t \ \le \ c_4 \left( \frac{2C}{n} \right)^2 \exp\big(c_1 L_\ell^2 \|\eta\|_2^2\big) \sum_{s=0}^{T-1} \eta_s^2. \qquad (11)$$

*Proof.* From $\Delta_{t+1} = \Delta_t - \eta_t \delta g_t$ and Lemma 3,

$$\mathbb{E}\|\Delta_{t+1}\|^2 \le \mathbb{E}\|\Delta_t\|^2 + \eta_t^2 \mathbb{E}\|\delta g_t\|^2 \le \mathbb{E}\|\Delta_t\|^2 + 2L^2 \eta_t^2 \mathbb{E}\|\Delta_t\|^2 + 2\eta_t^2 \left( \frac{2C}{b} \right)^2 q,$$

which is equation 9. Iterating the linear recursion and using $q = b/n$ gives equation 10–equation 11. $\qquad \square$

**Lemma 5** (Velocity discrepancy energy). *With $\delta v_t = \eta_t \delta g_t$,*

$$\mathbb{E} \sum_{t=0}^{T-1} \|\delta v_t\|^2 \ \le \ c_5 \left( \frac{2C}{n} \right)^2 e^{c_1 L^2 \|\eta\|_2^2} \sum_t \eta_t^2 \ + \ c_6 \left( \frac{2C}{n} \right)^2 \frac{\sum_t \eta_t^2}{q}. \qquad (12)$$

*Proof.* Use Lemma 3, $(a+b)^2 \le 2a^2 + 2b^2$, take expectations and sum; then apply equation 10 and $q = b/n$. $\qquad \square$

**Lemma 6** (Representation in the dominating LPK RKHS). *Let $\Delta \ell_T(z) := \ell(w_D(T); z) - \ell(w_{D'}(T); z)$ and write $\phi_s^D(z) := \nabla_w \ell(w_D(s); z)$, $\phi_s^{D'}(z) := \nabla_w \ell(w_{D'}(s); z)$. In the direct-sum RKHS $\mathcal{H}(K_{\ell,T}^\oplus)$,*

$$\Delta \ell_T(z) = \int_0^T \langle \phi_s^D(z), \dot{w}_D(s) \rangle \, ds \ + \ \int_0^T \langle -\phi_s^{D'}(z), \dot{w}_{D'}(s) \rangle \, ds,$$

*and therefore*

$$\|\Delta \ell_T\|_{\mathcal{H}(K_{\ell,T}^\oplus)} \ \le \ \sqrt{E_T^\oplus}. \qquad (13)$$

*Moreover, with $\bar{v} := \frac{1}{2}(\dot{w}_D + \dot{w}_{D'})$ and $\delta v := \dot{w}_D - \dot{w}_{D'}$,*

$$\Delta \ell_T = \frac{1}{2} \int_0^T \langle \phi_s^D + \phi_s^{D'}, \delta v(s) \rangle \, \mathrm{d}s \ + \ \frac{1}{2} \int_0^T \langle \phi_s^D - \phi_s^{D'}, \bar{v}(s) \rangle \, \mathrm{d}s, \qquad (14)$$

*and hence, using Assumption 3 for $\nabla_w \ell$,*

$$\|\Delta\ell_T\|_{\mathcal{H}(K_{\ell,T}^{\oplus})} \leq \left(\int_0^T \|\delta v(s)\|^2 \mathrm{d}s\right)^{1/2} + L_\ell \left(\int_0^T \|\Delta w(s)\|^2 \mathrm{d}s\right)^{1/2} \left(\frac{E_T^{\oplus}}{2}\right)^{1/2}. \tag{15}$$

*Proof.* By the reproducing property for $\mathcal{H}(K_{\ell,T}^{\oplus})$ and the feature-embedding identity equation 8 with features $\phi^D, \phi^{D'}$, we have the pathwise representation

$$\ell(w_D(T); z) - \ell(w_D(0); z) = \int_0^T \langle \phi_s^D(z), \dot{w}_D(s) \rangle \, \mathrm{d}s,$$

$$\ell(w_{D'}(T); z) - \ell(w_{D'}(0); z) = \int_0^T \langle \phi_s^{D'}(z), \dot{w}_{D'}(s) \rangle \, \mathrm{d}s.$$

Subtracting the two and embedding both terms in the direct sum yields the first display. Applying Cauchy–Schwarz in $\mathcal{H}(K_{\ell,T}^{\oplus})$ gives equation 13. For equation 14, rewrite the sum/difference using $\bar{v}$ and $\delta v$. Finally, Assumption 3 implies $\|\phi_s^D - \phi_s^{D'}\| \leq L_\ell \|\Delta w(s)\|$, and another Cauchy–Schwarz in time yields equation 15. $\qquad\square$

**Lemma 7** (Bounding the parameter-difference energy). *Let $E_T^{\Delta} := \int_0^T \|\Delta w(s)\|^2 \mathrm{d}s$. Then*

$$\mathbb{E}\, E_T^{\Delta} \leq c_7 \left(\frac{2C}{n}\right)^2 e^{c_1 L^2 \|\eta\|_2^2} \sum_t \eta_t^2 + c_8 \left(\frac{2C}{n}\right)^2 \frac{\sum_t \eta_t^2}{q}. \tag{16}$$

*Proof.* On $[t, t+1)$, $\Delta w(s) = \Delta_t - (s-t)\delta v_t$; integrate the square and sum, then use equation 11 and equation 12. $\qquad\square$

**Conclusion of the proof of Proposition 1.** Combine equation 15, Lemmas 5–7, and $\sqrt{a+b} \leq \sqrt{a} + \sqrt{b}$:

$$\mathbb{E}\, \|\Delta\ell_T\|_{\mathcal{H}(K_{\ell,T}^{\oplus})} \leq c_9 \left[\left(\mathbb{E} \sum_t \|\delta v_t\|^2\right)^{1/2} + L_g \sqrt{\frac{E_T^{\oplus}}{2}} \left(\mathbb{E}\, E_T^{\Delta}\right)^{1/2}\right].$$

Plugging equation 12–equation 16 yields, up to absolute constants,

$$\mathbb{E}\, \|\Delta\ell_T\|_{\mathcal{H}(K_{\ell,T}^{\oplus})} \leq \frac{2C}{n} \left(\sqrt{E_T^{\oplus}}\, \|\eta\|_2\, e^{c_1 L^2 \|\eta\|_2^2} + \sqrt{\frac{\sum_t \eta_t^2}{q}}\right),$$

which is the displayed bound in Proposition 1 for fixed $q$. For fixed $b$, use $q = b/n$ so that $\sqrt{(\sum_t \eta_t^2)/q} = \sqrt{n/b}\, \|\eta\|_2$, and factor out $1/\sqrt{n}$ to obtain $\widetilde{B}_T^{(b)}$. $\qquad\square$

### C.2 Proofs for the one-shot Gaussianization bounds

#### C.2.1 Proof of Proposition 2.

**Proposition** (Generic bound via second-order Poincaré). *Let $Z \sim \mathcal{N}(0, I_d)$ and $Y = g(Z)$ with $\sigma^2 := \mathrm{Var}(Y) > 0$. Define*

$$L_x := \left(\mathbb{E}\|\nabla_z g(Z)\|^4\right)^{1/4}, \qquad H_x := \left(\mathbb{E}\|\nabla_z^2 g(Z)\|_{\mathrm{HS}}^4\right)^{1/4}.$$

*Then*

$$d_{\mathrm{TV}}\big(\mathcal{L}(Y), \mathcal{N}(\mathbb{E}Y, \sigma^2)\big) = O\!\Big(\frac{L_x H_x}{\sigma^2}\Big).$$

*In particular, if $L_x = O(1)$ and $H_x = O(d_{\mathrm{eff}}^{-\beta})$ for some $\beta \in (0, 1]$ and $\sigma^2 = \Theta(d_{\mathrm{eff}})$, then $d_{\mathrm{TV}} = O(d_{\mathrm{eff}}^{-(1+\beta)})$.*

*Proof.* Let $G = (Y - \mathbb{E}Y)/\sigma$ with $\sigma^2 = \text{Var}(Y) > 0$. By the Gaussian second-order Poincaré inequality (TV version), see (Peccati et al., 2010, Eq. (1.6)),

$$d_{\text{TV}}\big(\mathcal{L}(G), \mathcal{N}(0,1)\big) \; \leq \; c\left(\mathbb{E}\|D^2 G\|_{\text{op}}^4\right)^{1/4}\left(\mathbb{E}\|DG\|^4\right)^{1/4}.$$

Since $DG = (\nabla g(Z))/\sigma$ and $D^2 G = (\nabla^2 g(Z))/\sigma$, and $\|\cdot\|_{\text{op}} \leq \|\cdot\|_{\text{HS}}$, we get

$$d_{\text{TV}}\big(\mathcal{L}(G), \mathcal{N}(0,1)\big) \; \leq \; c\,\frac{1}{\sigma^2}\left(\mathbb{E}\|\nabla^2 g(Z)\|_{\text{HS}}^4\right)^{1/4}\left(\mathbb{E}\|\nabla g(Z)\|^4\right)^{1/4} \; = \; O\Big(\frac{H_x L_x}{\sigma^2}\Big).$$

Since $d_{\text{TV}}(\mathcal{L}(Y), \mathcal{N}(\mathbb{E}Y, \sigma^2)) = d_{\text{TV}}(\mathcal{L}(G), \mathcal{N}(0,1))$, the claim follows. If moreover $L_x = O(1)$, $H_x = O(d_{\text{eff}}^{-\beta})$ and $\sigma^2 = \Theta(d_{\text{eff}})$, then $d_{\text{TV}} = O(d_{\text{eff}}^{-(1+\beta)})$. $\qquad\square$

### C.2.2 PROOF OF PROPOSITION 3.

**Proposition** (Variance–mixture bound for conditionally Gaussian outputs). *Suppose that, for some auxiliary sampler state $S$, the release is conditionally Gaussian with $A = \sigma^2(S) \geq 0$:*

$$Y \mid S \sim \mathcal{N}(0, A), \qquad \sigma^2 := \text{Var}(Y) = \mathbb{E}[A] > 0.$$

*Then*

$$d_{\text{TV}}\big(\mathcal{L}(Y), \mathcal{N}(0, \sigma^2)\big) \; \leq \; \frac{8\,\text{Var}(A)}{\sigma^4} \; = \; O\Big(\frac{\text{Var}(A)}{\sigma^4}\Big).$$

*In particular, if $\text{Var}(Y) = \Theta(d_{\text{eff}})$ and $\text{Var}(A) = O(d_{\text{eff}})$, then $d_{\text{TV}} = O(d_{\text{eff}}^{-1})$.*

*Proof.* Let $F := Y$ and $Z_\sigma \sim \mathcal{N}(0, \sigma^2)$ with $\sigma^2 := \text{Var}(Y) = \mathbb{E}[A] > 0$. Under $Y \mid S \sim \mathcal{N}(0, A)$ with $A \in L^2$, Proposition 5.4 of Favaro et al. (2025) gives

$$d_{\text{TV}}\big(\mathcal{L}(F), \mathcal{N}(0, \sigma^2)\big) \; \leq \; \frac{8\,\text{Var}(A)}{\sigma^4}.$$

The $O\big(d_{\text{eff}}^{-1}\big)$ rate follows once $\text{Var}(Y) = \Theta(d_{\text{eff}})$ and $\text{Var}(A) = O(d_{\text{eff}})$. $\qquad\square$

### C.2.3 INTERPRETATION OF THE VARIANCE–MIXTURE PARAMETER $A$

In Proposition 3, the release $Y$ is assumed to be conditionally Gaussian given an auxiliary state $S$, with random variance $A = \sigma^2(S)$. This setting arises naturally in many generative pipelines:

- **Variational autoencoders (VAE).** A decoder typically outputs both a mean and a variance parameter $\sigma^2(z)$ for latent input $z$. When we consider a scalar score $Y = s(x)$ of the generated sample $x$, the conditional distribution satisfies $Y \mid z \sim \mathcal{N}(\mu(z), \sigma^2(z))$, and here $A = \sigma^2(z)$.

- **Diffusion models.** Each reverse step injects Gaussian noise, $x_{t-1} = g_\theta(x_t, t) + \sigma_t \epsilon_t$, with $\epsilon_t \sim \mathcal{N}(0, I)$. A final scalar score $Y$ (e.g. a projection of $x_0$) is a linear combination of these injected noises, so that $Y \mid \{\epsilon_t\} \sim \mathcal{N}(0, A)$ with $A$ determined by the noise trajectory and coefficients along the path.

- **Temperature sampling.** Sampling from $\text{softmax}(f_\theta(x)/\tau)$ is equivalent to perturbing logits with random noise whose variance depends on the temperature $\tau$. For a scalar score derived from the sampled label, we can write $Y \mid \tau \sim \mathcal{N}(0, \tau^2 \sigma_{\text{base}}^2)$, hence $A = \tau^2 \sigma_{\text{base}}^2$.

Thus, $A$ represents the conditional variance contributed by stochastic elements in the generation process. The variance–mixture bound of Proposition 3 exploits this structure: instead of controlling high-order derivatives as in the Poincaré approach, it directly relates the Gaussian approximation error to $\text{Var}(A)$, which is often easier to estimate or bound in practice.

### C.2.4 PROOF OF LEMMA 2.

**Lemma 8** (Equal-variance shift between Gaussians). *Let $\tilde{P} = \mathcal{N}(m, \sigma_1^2)$, $\tilde{Q} = \mathcal{N}(m', \sigma_2^2)$ and $\bar{\sigma}^2 = \frac{1}{2}(\sigma_1^2 + \sigma_2^2)$. Define $\bar{P} = \mathcal{N}(m, \bar{\sigma}^2)$, $\bar{Q} = \mathcal{N}(m', \bar{\sigma}^2)$. Then there exists a universal $c_0 > 0$ such that*

$$d_{\text{TV}}(\tilde{P}, \bar{P}) + d_{\text{TV}}(\tilde{Q}, \bar{Q}) \; \leq \; c_0\,\frac{|\sigma_1^2 - \sigma_2^2|}{\bar{\sigma}^2}.$$

*Consequently, in our setting with $Y_D = f_D(Z), Y_{D'} = f_{D'}(Z)$, the equal-variance slack*

$$\omega := c_0 \frac{|\sigma_D^2 - \sigma_{D'}^2|}{\bar{\sigma}^2}$$

*satisfies*

$$\mathbb{E}[\omega] = O\left(\frac{1}{n\sqrt{C_T d_{\text{eff}}}}\right).$$

*Proof.* By Pinsker, $d_{\text{TV}}(P, Q) \leq \sqrt{\frac{1}{2} \text{KL}(P\|Q)}$. For equal-mean Gaussians,

$$\text{KL}(\mathcal{N}(m, \sigma_1^2) \| \mathcal{N}(m, \bar{\sigma}^2)) = \tfrac{1}{2}\left(\frac{\sigma_1^2}{\bar{\sigma}^2} - 1 - \log\frac{\sigma_1^2}{\bar{\sigma}^2}\right).$$

A Taylor expansion at $\sigma_1^2 = \bar{\sigma}^2$ yields $\text{KL} \leq C\left(\frac{\sigma_1^2 - \bar{\sigma}^2}{\bar{\sigma}^2}\right)^2$ for a universal $C$, hence $d_{\text{TV}}(\tilde{P}, \bar{P}) \leq c\left|\frac{\sigma_1^2 - \bar{\sigma}^2}{\bar{\sigma}^2}\right|$. Apply the same bound to $(\tilde{Q}, \bar{Q})$ and note $|\sigma_1^2 - \bar{\sigma}^2| + |\sigma_2^2 - \bar{\sigma}^2| = \frac{1}{2}|\sigma_1^2 - \sigma_2^2|$, which gives the displayed inequality for $c_0 = 2c$.

For the order estimate, observe that $\sigma_D^2 - \sigma_{D'}^2 = \mathbb{E}[f_D(Z)^2 - f_{D'}(Z)^2] = \mathbb{E}\langle f_D + f_{D'}, f_D - f_{D'}\rangle$. Cauchy–Schwarz in the common RKHS gives $|\sigma_D^2 - \sigma_{D'}^2| \lesssim \sqrt{\mathbb{E}\|f_D + f_{D'}\|^2} \|\Delta f\|_{\mathcal{H}}$. Assumption 2 bounds the first factor as $\Theta(\sqrt{C_T d_{\text{eff}}})$, while Proposition 1 yields $\mathbb{E}\|\Delta f\|_{\mathcal{H}} = O(1/n)$. Thus $\mathbb{E}|\sigma_D^2 - \sigma_{D'}^2| \lesssim \sqrt{C_T d_{\text{eff}}}/n$, and dividing by $\bar{\sigma}^2 = \Theta(C_T d_{\text{eff}})$ gives $\mathbb{E}[\omega] = O(1/(n\sqrt{C_T d_{\text{eff}}}))$. $\square$

### C.3 PROOF OF THEORY 1

Throughout this subsection we work in the RKHS of the dominating loss path kernel,

$$\mathcal{H} := \mathcal{H}(K_{\ell,T}^{\oplus}), \qquad K_{\ell,T}^{\oplus} := K_{\ell,T}^D + K_{\ell,T}^{D'},$$

and we write expectations with respect to $Z, Z' \overset{\text{i.i.d.}}{\sim} \mathcal{N}(0, I_d)$. Recall $Y_D = f_D(Z)$ and $Y_{D'} = f_{D'}(Z)$, and denote $\sigma_D^2 = \text{Var}(Y_D), \sigma_{D'}^2 = \text{Var}(Y_{D'})$.

We record the kernel mean embedding and the integral-operator bridge; both are standard and stated here for completeness.

**Lemma 9** (Kernel mean embedding: bridging mean differences)**.** *Let $\mathcal{H} = \mathcal{H}(K_T)$ be the RKHS of a positive definite kernel $K_T$, and define*

$$\mu(\cdot) := \mathbb{E}\big[K_T(\cdot, Z)\big] \in \mathcal{H}, \qquad Z \sim \mathcal{N}(0, I_d).$$

*Then for any $g \in \mathcal{H}$,*

$$\mathbb{E}\, g(Z) = \langle g, \mu\rangle_{\mathcal{H}}, \qquad \|\mu\|_{\mathcal{H}}^2 = \mathbb{E}_{Z,Z'} K_T(Z, Z').$$

*In particular, for any $f_D, f_{D'} \in \mathcal{H}, \big|\mathbb{E}\, f_D(Z) - \mathbb{E}\, f_{D'}(Z)\big| \leq \|f_D - f_{D'}\|_{\mathcal{H}} \|\mu\|_{\mathcal{H}}.$*

**Lemma 10** (Integral operator: bridging second-moment differences)**.** *Let $\mathcal{H} = \mathcal{H}(K_T)$ and define $\mathcal{T} : \mathcal{H} \to \mathcal{H}$ by*

$$\langle g, \mathcal{T}h\rangle_{\mathcal{H}} := \mathbb{E}\big[g(Z)\, h(Z)\big].$$

*Then for any $f_D, f_{D'} \in \mathcal{H}$,*

$$\big|\mathbb{E}\, f_D(Z)^2 - \mathbb{E}\, f_{D'}(Z)^2\big| = \big|\langle f_D + f_{D'}, \mathcal{T}(f_D - f_{D'})\rangle_{\mathcal{H}}\big| \leq \big(\|f_D\|_{\mathcal{H}} + \|f_{D'}\|_{\mathcal{H}}\big) \|\mathcal{T}\|_{\text{op}} \|f_D - f_{D'}\|_{\mathcal{H}},$$

*and moreover $\|\mathcal{T}\|_{\text{op}} \leq \mathbb{E}\, K_T(Z, Z).$*

We now prove Theorem 1. Define the Gaussian approximations $\tilde{P} = \mathcal{N}(\mathbb{E}Y_D, \sigma_D^2)$ and $\tilde{Q} = \mathcal{N}(\mathbb{E}Y_{D'}, \sigma_{D'}^2)$.

*Proof of Theorem 1.* By Proposition 2 and Proposition 3 , set

$$\gamma_d := \max\Big\{d_{\text{TV}}\big((Y_D - \mathbb{E}Y_D)/\sigma_D, \mathcal{N}(0, 1)\big), d_{\text{TV}}\big((Y_{D'} - \mathbb{E}Y_{D'})/\sigma_{D'}, \mathcal{N}(0, 1)\big)\Big\},$$

so that $d_{\text{TV}}(P_{Y_D}, \tilde{P}) \leq \gamma_d$ and $d_{\text{TV}}(P_{Y_{D'}}, \tilde{Q}) \leq \gamma_d$.

*Step 1 (Gaussianization and robustness).* Applying the total-variation robustness of the tradeoff function (Lemma 1) with $\varepsilon = \gamma_d$, for any $\alpha \in (\gamma_d, 1 - \gamma_d)$,

$$T(P_{Y_D}, P_{Y_{D'}})(\alpha) \geq T(\tilde{P}, \tilde{Q})(\alpha + \gamma_d) - \gamma_d. \tag{17}$$

*Step 2 (Equalizing variances).* Let $\bar{\sigma}^2 = \frac{1}{2}(\sigma_D^2 + \sigma_{D'}^2)$ and define $\bar{P} = \mathcal{N}(\mathbb{E}Y_D, \bar{\sigma}^2)$, $\bar{Q} = \mathcal{N}(\mathbb{E}Y_{D'}, \bar{\sigma}^2)$. Using the equal-mean Gaussian TV bound (Pinsker–KL or a direct TV/Hellinger estimate), there exists an absolute constant $C$ such that

$$d_{\mathrm{TV}}(\tilde{P}, \bar{P}) \leq C \frac{|\sigma_D^2 - \bar{\sigma}^2|}{\bar{\sigma}^2}, \qquad d_{\mathrm{TV}}(\tilde{Q}, \bar{Q}) \leq C \frac{|\sigma_{D'}^2 - \bar{\sigma}^2|}{\bar{\sigma}^2}.$$

Write $\omega := C \frac{|\sigma_D^2 - \bar{\sigma}^2| + |\sigma_{D'}^2 - \bar{\sigma}^2|}{\bar{\sigma}^2}$. A second application of Lemma 1 gives, for all $\alpha \in (\gamma_d + \omega, 1 - \gamma_d - \omega)$,

$$T(\tilde{P}, \tilde{Q})(\alpha + \gamma_d) \geq T(\bar{P}, \bar{Q})(\alpha + \gamma_d + \omega) - \omega. \tag{18}$$

Combining equation 17–equation 18,

$$T(P_{Y_D}, P_{Y_{D'}})(\alpha) \geq T(\bar{P}, \bar{Q})(\alpha + \gamma_d + \omega) - (\gamma_d + \omega). \tag{19}$$

*Step 3 (Equal-variance Gaussian pairs yield GDP).* For $\bar{P} = \mathcal{N}(m_1, \bar{\sigma}^2)$ and $\bar{Q} = \mathcal{N}(m_2, \bar{\sigma}^2)$, the Neyman–Pearson tradeoff equals $G_\mu$ with $\mu = |m_1 - m_2|/\bar{\sigma}$. Hence

$$T(P_{Y_D}, P_{Y_{D'}})(\alpha) \geq G_\mu(\alpha + \gamma_d + \omega) - (\gamma_d + \omega), \tag{20}$$

where $\mu = \frac{|\mathbb{E}Y_D - \mathbb{E}Y_{D'}|}{\bar{\sigma}}$.

*Step 4 (Bounding $\mu$ in $\mathcal{H}(K_{\ell,T}^\oplus)$).* By Lemma 9 applied with $K_T = K_{\ell,T}^\oplus$,

$$|\mathbb{E}Y_D - \mathbb{E}Y_{D'}| = |\langle f_D - f_{D'}, \mu_K \rangle_\mathcal{H}| \leq \|\Delta f\|_\mathcal{H} \|\mu_K\|_\mathcal{H}, \quad \|\mu_K\|_\mathcal{H}^2 = \mathbb{E}_{Z,Z'} K_{\ell,T}^\oplus(Z, Z').$$

Under Assumption 2, $\mathbb{E}_{Z,Z'} K_{\ell,T}^\oplus(Z, Z') = \Theta(C_T)$ and $\bar{\sigma}^2 = \Theta(\mathbb{E} K_{\ell,T}^\oplus(Z, Z)) = \Theta(C_T d)$. Therefore

$$\mu \lesssim \frac{\|\Delta f\|_{\mathcal{H}(K_{\ell,T}^\oplus)}}{\sqrt{\mathbb{E} K_{\ell,T}^\oplus(Z, Z)}}. \tag{21}$$

*Step 5 (Bounding the equalization slack $\omega$).* By Lemma 10 in $\mathcal{H} = \mathcal{H}(K_{\ell,T}^\oplus)$,

$$|\sigma_D^2 - \sigma_{D'}^2| \leq (\|f_D\|_\mathcal{H} + \|f_{D'}\|_\mathcal{H}) \|\mathcal{T}\|_{\mathrm{op}} \|\Delta f\|_\mathcal{H}, \qquad \|\mathcal{T}\|_{\mathrm{op}} \leq \mathbb{E} K_{\ell,T}^\oplus(Z, Z).$$

Since $|\sigma_D^2 - \bar{\sigma}^2| + |\sigma_{D'}^2 - \bar{\sigma}^2| = \frac{1}{2} |\sigma_D^2 - \sigma_{D'}^2|$,

$$\omega \lesssim \frac{(\|f_D\|_\mathcal{H} + \|f_{D'}\|_\mathcal{H}) \|\Delta f\|_\mathcal{H}}{\mathbb{E} K_{\ell,T}^\oplus(Z, Z)}. \tag{22}$$

Note that $\omega$ appears only as a horizontal slack; it does not need to be absorbed into $\mu$.

*Step 6 (Conclusion and rates).* Because $G_\mu(\cdot)$ is 1-Lipschitz in its argument and nondecreasing in $\mu$, equation 20 together with equation 21 implies the stated GDP lower bound with

$$\mu_{\mathrm{eff}} \lesssim \frac{\|\Delta f\|_{\mathcal{H}(K_{\ell,T}^\oplus)}}{\sqrt{\mathbb{E} K_{\ell,T}^\oplus(Z, Z)}}.$$

Taking expectation over the DP–SGD randomness and invoking Proposition 1 (path-kernel stability under clipping, noise and subsampling),

$$\mathbb{E} \|\Delta f\|_{\mathcal{H}(K_{\ell,T}^\oplus)} = O(1/n)$$

and using Assumption 2, $\mathbb{E} K_{\ell,T}^\oplus(Z, Z) = \Theta(C_T d)$, we obtain

$$\mathbb{E} \mu_{\mathrm{eff}} = \mathcal{O}\left(\frac{1}{n \sqrt{C_T d_{\mathrm{eff}}}}\right)$$

with admissible $\alpha \in (\gamma_d + \omega, 1 - \gamma_d - \omega)$ as in equation 19. This completes the proof. $\qquad\square$

## C.4    Technical details for multi-release analysis

This appendix supplies the technical ingredients used in §4.5, keeping the presentation close to the single-release tools in §4.3 and App. C.3.

**TV–Hellinger relations.**    For any probability measures $P, Q$ on a common measurable space, with $H^2(P, Q) := \frac{1}{2}\int(\sqrt{dP} - \sqrt{dQ})^2$,

$$H^2(P, Q) \ \leq \ d_{\mathrm{TV}}(P, Q) \ \leq \ \sqrt{2}\, H(P, Q). \tag{23}$$

**Hellinger tensorization for i.i.d. products.**    For $m \geq 1$,

$$H^2\big(P^{\otimes m}, Q^{\otimes m}\big) \ = \ 1 - \big(1 - H^2(P, Q)\big)^m. \tag{24}$$

In particular, when $H(P, Q) \ll 1$, $H\big(P^{\otimes m}, Q^{\otimes m}\big) \leq \sqrt{m}\, H(P, Q)$.

**Equal-variance shift between Gaussians.**    We reuse Lemma 8 (App. C.3): for $\tilde{P} = \mathcal{N}(m, \sigma_1^2)$ and $\bar{P} = \mathcal{N}(m, \bar{\sigma}^2)$ with $\bar{\sigma}^2 = (\sigma_1^2 + \sigma_2^2)/2$, there is a universal $c_0 > 0$ such that $d_{\mathrm{TV}}(\tilde{P}, \bar{P}) \leq c_0\, |\sigma_1^2 - \bar{\sigma}^2|/\bar{\sigma}^2$ (and symmetrically for $Q$).

**Product-level Gaussianization & equal-variance slack.**    Let $P_{Y_D}, P_{Y_{D'}}$ be the true one-release laws, $\tilde{P}, \tilde{Q}$ their Gaussian surrogates, and $\bar{P}, \bar{Q}$ the equal-variance Gaussians. Write $\varepsilon := \gamma_d + \omega$, where

$$\gamma_d \ := \ \max\Big\{ d_{\mathrm{TV}}\big((Y_D - \mathbb{E}Y_D)/\sigma_D, \mathcal{N}(0, 1)\big), \ d_{\mathrm{TV}}\big((Y_{D'} - \mathbb{E}Y_{D'})/\sigma_{D'}, \mathcal{N}(0, 1)\big) \Big\},$$

and $\omega$ is the equal-variance slack from equation 3. We prove the bound used in equation 5.

**Lemma 11** (Growth of the product-level slack)**.** *There exists an absolute constant $c > 0$ such that*

$$\begin{aligned} \Gamma_m \ :=&\ d_{\mathrm{TV}}\big(P_{Y_D}^{\otimes m}, \ \tilde{P}^{\otimes m}\big) + d_{\mathrm{TV}}\big(P_{Y_{D'}}^{\otimes m}, \ \tilde{Q}^{\otimes m}\big) + d_{\mathrm{TV}}\big(\tilde{P}^{\otimes m}, \ \bar{P}^{\otimes m}\big) + d_{\mathrm{TV}}\big(\tilde{Q}^{\otimes m}, \ \bar{Q}^{\otimes m}\big) \\ &\leq \ c\, \min\{\, m\varepsilon, \ 2\sqrt{m\varepsilon}\,\}. \end{aligned}$$

*Proof Sketch. Hybrid (linear) bound.* By triangle inequality and a replace-one-coordinate (hybrid) argument, $d_{\mathrm{TV}}\big(P^{\otimes m}, Q^{\otimes m}\big) \leq m\, d_{\mathrm{TV}}(P, Q)$ for any pair $(P, Q)$. Summing the four terms yields $\Gamma_m \leq c\, m\varepsilon$.

*Hellinger (square-root) bound.* For each of the four terms, apply equation 23 and equation 24:

$$d_{\mathrm{TV}}\big(P^{\otimes m}, Q^{\otimes m}\big) \ \leq \ \sqrt{2}\, H\big(P^{\otimes m}, Q^{\otimes m}\big) \ \leq \ \sqrt{2m}\, H(P, Q) \ \leq \ 2\sqrt{m}\, d_{\mathrm{TV}}(P, Q).$$

Again summing the four contributions gives $\Gamma_m \leq c\, 2\sqrt{m\,\varepsilon}$. Taking the minimum of the two completes the proof. $\square$

**Product-TV sandwich (signal side).**    For completeness we recall a standard sandwich bound that we use only as a tool to reason about *signal* accumulation under products:

**Theorem 3** (Product-TV sandwich, e.g. Kontorovich (2025); Polyanskiy & Wu (2025))**.** *Let $\delta = d_{\mathrm{TV}}(P, Q)$. Then there exist absolute constants $0 < c \leq C < \infty$ such that*

$$\max\Big\{ c\, \min\{1, \sqrt{m}\, \delta\}, \ \Theta\big(\min\{1, m\delta^2\}\big) \Big\} \ \leq \ d_{\mathrm{TV}}\big(P^{\otimes m}, Q^{\otimes m}\big) \ \leq \ C\, \min\{1, \sqrt{m}\, \delta\}.$$

This theorem explains the $\sqrt{m}$ accumulation of the Neyman–Pearson signal in the *Gaussian surrogate*, which we express in GDP form as $\mu_{\mathrm{eff}}^{(m)} = \sqrt{m}\, \mu_{\mathrm{eff}}^{(1)}$.

**Putting the pieces together.**    Lemma 11 controls the product-level slack $\Gamma_m$ that shifts the tradeoff; Theorem 3 controls the signal-side growth and yields the $\sqrt{m}$ law for $\mu_{\mathrm{eff}}$ in the Gaussian world. A final application of the TV-robustness of the tradeoff function (Lemma 1) delivers the main-text bound equation 6.

**Guide for audits/plots.** When instantiating the envelope, we recommend using the explicit slack $\Gamma_m = \min\{ m\varepsilon,\ 2\sqrt{m\varepsilon}\}$ with $\varepsilon = \gamma_d + \omega$ kept *outside* of big-O notation, to avoid scale confusion between $m$ and $\sqrt{m}$. For typical $\gamma_d = O(1/d_{\text{eff}})$ and negligible $\omega$, this reduces to $\Gamma_m = O\big( \min\{m/d_{\text{eff}},\ \sqrt{m/d_{\text{eff}}}\}\big)$, matching equation 7 in the main text.

# D IMPLEMENTATION DETAILS AND APPLICATIONS

## D.1 PRACTICAL TRACKING OF GDP CERTIFICATES WITH LPK

This section specifies how we estimate the LPK statistics that enter Theorem 1 and the stability bound of §4.2 from a *single* DP–SGD trajectory. Let $w_0, \ldots, w_T$ be the parameter path, with step sizes $\{\eta_t\}$. Let $Z, Z' \overset{\text{i.i.d.}}{\sim} \mathcal{N}(0, I_d)$ be independent of the training data. We fix a smooth calibration loss $\ell_{\text{cal}}(w, z)$ with bounded output derivative and define the loss path kernel

$$K_{\ell,T}(z, z') := \int_0^T \big\langle \nabla_w \ell_{\text{cal}}(w(t), z),\ \nabla_w \ell_{\text{cal}}(w(t), z') \big\rangle \, \mathrm{d}t,$$

which we discretize as $K_{\ell,T}(z, z') \approx \sum_{t=0}^{T-1} \eta_t \langle g_t(z), g_t(z') \rangle$ with $g_t(z) := \nabla_w \ell_{\text{cal}}(w_t, z)$. Throughout, expectations over $Z$ are with respect to the dominating kernel $K_{\ell,T}^{\oplus}$; in the one-run certificate we approximate $\mathbb{E}K_{\ell,T}^{\oplus} \approx 2\,\mathbb{E}K_{\ell,T}$ by symmetry of the coupled execution.

### D.1.1 ESTIMATING KERNEL SCALES

Draw an i.i.d. probe set $\{Z^{(m)}\}_{m=1}^M$ once and reuse it for all $t$. For each $t$, compute $g_t(Z^{(m)})$ and cache on CPU (no optimizer state). Define Monte Carlo estimates:

$$\widehat{K}_{\ell,T}(Z^{(m)}, Z^{(m)}) := \sum_{t=0}^{T-1} \eta_t \, \|g_t(Z^{(m)})\|^2,$$

$$\widehat{K}_{\ell,T}(Z^{(m)}, Z^{(m')}) := \sum_{t=0}^{T-1} \eta_t \, \langle g_t(Z^{(m)}), g_t(Z^{(m')})\rangle,$$

$$\widehat{\mathbb{E}K_{\ell,T}}(Z, Z) := \frac{1}{M} \sum_{m=1}^M \widehat{K}_{\ell,T}(Z^{(m)}, Z^{(m)}),$$

$$\widehat{\mathbb{E}K_{\ell,T}}(Z, Z') := \frac{1}{M(M-1)} \sum_{m \neq m'} \widehat{K}_{\ell,T}(Z^{(m)}, Z^{(m')}).$$

In the common-RKHS convention we use $\widehat{\mathbb{E}K_{\ell,T}^{\oplus}}(Z, Z) \approx 2\,\widehat{\mathbb{E}K_{\ell,T}}(Z, Z)$ and $\widehat{\mathbb{E}K_{\ell,T}^{\oplus}}(Z, Z') \approx 2\,\widehat{\mathbb{E}K_{\ell,T}}(Z, Z')$.

### D.1.2 TRACKING THE STABILITY SIDE FROM ONE RUN

Proposition 1 upper-bounds $\mathbb{E}\|\Delta f\|_{\mathcal{H}(K_{\ell,T}^{\oplus})}$ by a function of clipping $C$, noise scale $\sigma$, subsampling rate $q = b/n$, and the schedule $\{\eta_t\}$, without requiring paired trainings at run time. We therefore form

$$\widehat{B}_T := 2C \left( \sqrt{\widehat{E}_T^{\oplus}} \, \|\eta\|_2 \, e^{\, c_1 L_\ell^2 \|\eta\|_2^2} \ + \ \sqrt{\frac{\sum_t \eta_t^2}{q}} \right), \quad \|\eta\|_2 = \left( \sum_t \eta_t^2 \right)^{1/2},$$

where $\widehat{E}_T^{\oplus}$ is a path-energy proxy recorded from the run (e.g. $\widehat{E}_T^{\oplus} \approx 2 \sum_t \eta_t^2 \|g_t(Z)\|^2$ averaged over probes), and $c_1$ is the absolute constant from the proof. When $\sum_t \eta_t^2 = O(1)$ the exponential factor is $O(1)$.

### D.1.3 ESTIMATING GAUSSIANIZATION ERROR AND EQUAL-VARIANCE SLACK

For a scalar release $Y = f_{w_T}(Z)$ we measure the Gaussianization error via a projection-based proxy. Generate a fresh probe set $\{W^{(r)}\}_{r=1}^R$ with $W^{(r)} \overset{\text{i.i.d.}}{\sim} \mathcal{N}(0, I_d)$. For vector outputs, draw $u_1, \ldots, u_S$ uniformly on the unit sphere in the output space and set $Y_s = \langle u_s, f_{w_T}(W) \rangle$; for scalar outputs take $S = 1$. Define

$$\widehat{\gamma} := \max_{1 \leq s \leq S} d_{\mathrm{KS}}\Big(\frac{Y_s - \bar{Y}_s}{\widehat{\sigma}_s}, \mathcal{N}(0,1)\Big),$$

where $d_{\mathrm{KS}}$ is the Kolmogorov–Smirnov distance, $\bar{Y}_s$ and $\widehat{\sigma}_s^2$ are the empirical mean/variance. This estimates the $\gamma_d$ in §4.3. For the equal-variance slack, let $\sigma_D^2$ and $\sigma_{D'}^2$ be the empirical variances of $Y_D$ and $Y_{D'}$ under the same probe seed; set

$$\widehat{\omega} := c_0 \frac{|\sigma_D^2 - \sigma_{D'}^2|}{(\sigma_D^2 + \sigma_{D'}^2)/2}.$$

In the one-run certificate, we replace $\widehat{\omega}$ by the plug-in bound from the proof of Lemma 2, which yields $\mathbb{E}[\omega] \lesssim (\|\Delta f\|_{\mathcal{H}}/\sqrt{\mathbb{E}K}) \cdot (\|f_D\|_{\mathcal{H}} + \|f_{D'}\|_{\mathcal{H}})/\mathbb{E}K$. With $\|\Delta f\|_{\mathcal{H}}$ replaced by $\widehat{B}_T$ and $\mathbb{E}K$ by $\widehat{\mathbb{E}K_{\ell,T}^{\oplus}}(Z,Z)$, we use

$$\widehat{\omega}_{\text{1-run}} := c_0 \frac{\widehat{R}_T \, \widehat{B}_T}{\widehat{\mathbb{E}K_{\ell,T}^{\oplus}}(Z,Z)}, \qquad \widehat{R}_T := \frac{\|f_D\|_{\mathcal{H}} + \|f_{D'}\|_{\mathcal{H}}}{\sqrt{\widehat{\mathbb{E}K_{\ell,T}^{\oplus}}(Z,Z)}},$$

where $\|f_D\|_{\mathcal{H}}$ is estimated from the same probes via the standard kernel ridge formula (with a small ridge $\lambda$ for numerical stability).[1]

### D.1.4 ASSEMBLING THE CERTIFICATE

Given $\widehat{B}_T$, $\widehat{\mathbb{E}K_{\ell,T}^{\oplus}}(Z,Z)$ and $\widehat{\gamma}$, define the one-shot effective parameter

$$\widehat{\mu}_{\text{eff}} := \frac{\widehat{B}_T}{\sqrt{\widehat{\mathbb{E}K_{\ell,T}^{\oplus}}(Z,Z)}}.$$

The product-level slack for $m$ independent releases is $\widehat{\Gamma}_m := \min\{ m(\widehat{\gamma}+\widehat{\omega}), \, 2\sqrt{m(\widehat{\gamma}+\widehat{\omega})} \}$. The corresponding envelope is $G_{\sqrt{m}\,\widehat{\mu}_{\text{eff}}}(\alpha + \widehat{\Gamma}_m) - \widehat{\Gamma}_m$ on $\alpha \in (\widehat{\Gamma}_m, \, 1 - \widehat{\Gamma}_m)$.

### D.1.5 SIMULATION DETAILS USED IN §5

All figures are computed from a single DP–SGD run of a small MLP-based VAE trained on a standard image dataset.[2] We use a cosine learning-rate schedule, weight decay $10^{-4}$, clipping $C = 1$, fixed subsampling rate $q = b/n$, and Gaussian noise $\sigma$ as stated in the captions. For probe statistics we take $M \in \{16, 32, 64\}$ Gaussian probes, re-used across $t$, log every 20–40 steps, and set the ridge $\lambda = 10^{-6}$. For $\widehat{\gamma}$ we use projection count $S \in \{8, 16, 32\}$ and report the worst direction (KS);Cramér–von Mises gives consistent values and is used as a cross-check. For the $d_{\text{eff}}$ sweeps we vary the latent dimension while keeping architecture and training hyperparameters fixed. For the $n$ sweeps we hold $q$ fixed and scale $b$ with $n$.

### D.1.6 DIAGNOSTICS, SMOOTHING, AND COST

Numerical stability improves if we: (i) reuse the same probe set across $t$; (ii) enforce nonempty mini-batches under subsampling; (iii) apply a small ridge $\lambda$ to kernel matrices; (iv) median-smooth $\widehat{\mu}_{\text{eff}}$ across consecutive checkpoints before the final average. Computationally, the overhead is a first-order pass per checkpoint: $O\big(M \sum_t \text{cost}(\nabla_w \ell_{\text{cal}}(\cdot, Z))\big)$, typically $< 10\%$ wall-clock with $M \leq 64$.

---

[1]In practice we use $\lambda \in [10^{-6}, 10^{-5}]$ and check that estimates are insensitive to $\lambda$ within this range.

[2]Data details are immaterial for the certificate; we use the MNIST dataset (28×28 grayscale digits) for reproducibility.

### D.1.7 SANITY CHECKS

Near-isotropy can be diagnosed by checking that $\widehat{\mathbb{E}K_{\ell,T}}(Z,Z)$ grows approximately linearly with the latent dimension and that the ratio $\widehat{\mathbb{E}K_{\ell,T}}(Z,Z)/\widehat{\mathbb{E}K_{\ell,T}}(Z,Z')$ is roughly stable. If not, we normalize probes or increase $M$. For the slacks, $\widehat{\gamma}$ should decrease with effective dimension, and $\widehat{\omega}$ should shrink with $n$; large deviations usually indicate insufficient probes or early-epoch transients.

## D.2 APPLICATIONS TO COMMON ARCHITECTURES

This section discusses how the assumptions behind Theorem 1 instantiate in common generative architectures, and how the effective dimension $d_{\text{eff}}$ should be interpreted and estimated in each case. Throughout, near-isotropy is checked with the probe-based diagnostics of Appendix D.1, using the ratio

$$d_{\text{eff}} := \frac{\mathbb{E}K_{\ell,T}^{\oplus}(W,W)}{\mathbb{E}_{W,W'}K_{\ell,T}^{\oplus}(W,W')}, \qquad W, W' \overset{\text{i.i.d.}}{\sim} \Pi,$$

and the same calibration loss $\ell_{\text{cal}}$ as in the experiments.

### D.2.1 FULLY-CONNECTED DECODERS AND SHALLOW MLPs

For a width-$m$ MLP with smooth activations (softplus/gelu) and standard width–dimension scaling, Assumption 1 holds with bounded output derivatives and finite Gaussian gradients/Hessians; Assumption 3 follows from local Lipschitzness of the parameter-to-output Jacobian along the path. With random Gaussian probes $W \sim \mathcal{N}(0, I_d)$, the LPK scale concentrates and near-isotropy typically gives $\mathbb{E}K_{\ell,T}^{\oplus}(W,W) = \Theta(C_T d)$ and $\mathbb{E}_{W,W'}K_{\ell,T}^{\oplus}(W,W') = \Theta(C_T)$, so $d_{\text{eff}} \simeq d$. Under these controls, Proposition 2 yields $\gamma_d = \tilde{O}(d^{-1/2})$ (often faster when features decorrelate), and Theorem 1 implies $\mathbb{E}\mu_{\text{eff}} = O\big(1/(n\sqrt{d})\big)$ in the fixed-$q$ regime up to higher-order slacks.

### D.2.2 VARIATIONAL AUTOENCODERS (VAE) AND LATENT-GAUSSIAN DECODERS

When sampling uses $z \sim \mathcal{N}(0, I_{d_z})$ and a smooth decoder $x = g_\theta(z)$, the natural probe law is $\Pi = \mathcal{N}(0, I_{d_z})$ pushed through the score $\ell_{\text{cal}}(w,z)$. Assumption 1 reduces to smoothness of the decoder and bounded $\partial_f \ell_{\text{cal}}$; Assumption 3 follows from smoothness of $\nabla_w f$. The near-isotropy scale is typically governed by $d_z$, hence $d_{\text{eff}} \simeq d_z$. Because the release is often a variance mixture (conditioning on latent/temperature noise), Proposition 3 applies and gives $\gamma_d = O(1/d_{\text{eff}})$ under mild variance concentration, sharpening the Gaussianization side relative to the generic bound.

### D.2.3 DIFFUSION SAMPLERS

A $T$-step reverse diffusion with Gaussian injections admits a Gaussian probe $W = (x_T, \eta_T, \ldots, \eta_1) \sim \mathcal{N}(0, I_{d(T+1)})$. The effective dimension should account for the sensitivity of the final sample to the injected noises:

$$d_{\text{eff}} \approx d\left(1 + \sum_{t=1}^{T} \chi_t\right), \qquad \chi_t := \frac{1}{d}\mathbb{E}\left\|\frac{\partial x_0}{\partial \eta_t}\right\|_F^2,$$

so $d \leq d_{\text{eff}} \leq d(1+T)$, with equality on the left for deterministic schedules (e.g. DDIM). Near-isotropy holds when the noise schedule is well-conditioned and gradients along the path are not dominated by a narrow subset of timesteps; otherwise the anisotropy is revealed by an inflated on/off-diagonal ratio and can be mitigated by probe normalization. Conditional Gaussianity makes Proposition 3 natural here, giving $\gamma_d = O(1/d_{\text{eff}})$.

### D.2.4 NORMALIZING FLOWS

Flows preserve latent dimensionality; with $z \sim \mathcal{N}(0, I_{d_z})$ and smooth invertible maps, Assumptions 1 and 3 hold under standard Jacobian regularity. Near-isotropy and $d_{\text{eff}} \simeq d_z$ follow from the probe symmetry. Since sampling is deterministic given $z$, the generic second-order Poincaré bound (Proposition 2) is the appropriate Gaussianization tool; empirical calibration may be helpful at moderate $d_z$.

### D.2.5 AUTOREGRESSIVE TRANSFORMERS

We consider a decoder-only Transformer with randomized decoding (temperature or nucleus sampling). Let $x_{1:t-1}$ be the fixed context and $h_t$ the hidden state at step $t$. Randomness enters through the sampling mechanism. To align with our probe-based analysis, we model this by injecting independent Gaussian probes either (i) in the embedding stream at the sampled positions, or (ii) in the pre-softmax logits:

$$e_t \;\mapsto\; e_t + \sigma_{\mathrm{emb}}\xi_t, \quad \text{or} \quad \ell_t \;\mapsto\; \ell_t + \tau\,\xi_t, \qquad \xi_t \stackrel{\mathrm{i.i.d.}}{\sim} \mathcal{N}(0, I_{d_{\mathrm{emb}}}).$$

This "Gaussian surrogate" is standard for sensitivity and CLT analyses and closely approximates the variability introduced by multinomial/Gumbel sampling when $\sigma_{\mathrm{emb}}, \tau$ match the empirical variance.

*Assumption 1* (model and Gaussian regularity) holds because attention blocks (linear maps, softmax, layer norm, smooth activations) are differentiable and admit bounded local derivatives along the DP–SGD path; the required $\nabla_z g$ and $\nabla_z^2 g$ moments follow from spectral controls on the per-layer Jacobians. *Assumption 3* is met under the usual DP–SGD clipping plus weight decay, which bound the growth of parameter-to-output Lipschitz factors. For *Assumption 2*, near-isotropy can be checked empirically by comparing on/off–diagonal LPK scales under probes $\{\xi_t\}$; mild anisotropy can be mitigated by whitening probes with the estimated covariance of embedding directions.

The effective dimension is governed by the embedding width and the number of stochastic decoding steps:

$$d_{\mathrm{eff}} \;\approx\; d_{\mathrm{emb}} \sum_{t=1}^{m} \rho_t, \qquad \rho_t \in [0, 1] \text{ encodes the sensitivity of the score to the } t\text{-th probe,}$$

so $d_{\mathrm{emb}} \le d_{\mathrm{eff}} \le m\, d_{\mathrm{emb}}$. Empirically $\rho_t$ reflects attention spread and temperature: lower temperatures or very small top-$k$ reduce $\rho_t$ and hence $d_{\mathrm{eff}}$.

On the Gaussianization side, two routes apply. (a) With logit probes $\ell_t + \tau\xi_t$, conditional on the network state the scalarized release $Y$ is well-approximated by a Gaussian with variance proportional to $\sum_t \|\partial Y / \partial \ell_t\|^2$, yielding the variance–mixture bound $\gamma_d = O(1/d_{\mathrm{eff}})$ (Prop. 3). (b) With purely discrete sampling (no explicit Gaussian jitter), the second-order Gaussian Poincaré route (Prop. 2) applies to the smooth functional $g(\xi)$ obtained by a local linearization of the sampling step, again leading to $\gamma_d$ decaying with $d_{\mathrm{eff}}$ under standard smoothness.

For the LPK side, the per-step gradient features entering $K_{\ell,T}$ are the parameter gradients of a smooth calibration score built from intermediate or final hidden states (e.g., squared norm or a fixed linear probe). Along clipped, noisy DP–SGD, the resulting function-level discrepancy in the common RKHS obeys the same stability scaling as in the general theory; the kernel's expected on-diagonal scale grows like $C_T d_{\mathrm{eff}}$, while the off-diagonal stays at $\Theta(C_T)$.

Failure modes are informative: deterministic decoding (greedy/beam) effectively sets $d_{\mathrm{eff}} \approx 0$, eliminating amplification; highly anisotropic embeddings or attention collapse can violate near-isotropy, which should be diagnosed via probe statistics and addressed by whitening or by injecting small Gaussian logit noise during release. In typical temperature or nucleus sampling with modern decoders, the assumptions are well aligned and the predicted $d_{\mathrm{eff}}^{-1/2}$ envelope (or better) is observed.

### D.2.6 DIAGNOSTICS, FAILURE MODES, AND HOW TO PROCEED

Across architectures, the three assumptions admit concrete checks: (i) *Model and Gaussian regularity* (Assumption 1) reduces to verifying smooth activations, bounded $\partial_f \ell_{\mathrm{cal}}$, and finite Gaussian gradient/Hessian moments of the decoder w.r.t. probes; (ii) *Near-isotropic kernel scale* (Assumption 2) is diagnosed by comparing Monte Carlo estimates of $\mathbb{E} K_{\ell,T}^{\oplus}(W, W)$ and $\mathbb{E}_{W,W'} K_{\ell,T}^{\oplus}(W, W')$; (iii) *Lipschitz loss gradients* (Assumption 3) follow from path-wise bounds on $\|\nabla_w f\|$ under clipping and smooth blocks. If near-isotropy is violated (e.g. overly concentrated receptive fields, deterministic decoding, or attention collapse), one can (a) re-normalize probes to equalize variance across coordinates or timesteps, (b) switch to a calibration score that restores variance, or (c) report the conservative envelope with the empirically estimated $d_{\mathrm{eff}}$ as-is. In all cases, the certificate depends only on $C_T$, the probe-based kernel scales, and the LPK stability bound, and can therefore be tracked with the lightweight procedure of Appendix D.1.

### D.3 LIMITATIONS AND FUTURE WORK

This section discusses where our assumptions may weaken, why our bounds are conservative, how the framework may extend beyond DP–SGD, and what changes under stronger threat models.

#### D.3.1 ON ASSUMPTIONS

Our results hinge on Assumptions 1–3 and near-isotropy in Assumption 2. *Model regularity (Assumption 1).* While differentiability and local Lipschitzness are satisfied by common smooth activations (e.g., GELU/Softplus), hard nonlinearities (ReLU with sharp kinks, max-pooling) or aggressive normalization can inflate local Jacobians and Hessians, weakening the moment bounds that underlie the Gaussianization rates in §2.3. In practice, replacing nonsmooth units by smooth surrogates along the analysis (not necessarily in training) and using empirical probe diagnostics mitigate this issue. *Near-isotropy (Assumption 2).* Architectures with strong anisotropy (e.g., narrow bottlenecks, collapsed attention heads, rank-deficient upsamplers) may yield $\mathbb{E}K_{\ell,T}^{\oplus}(W,W) \ll C_T d_{\mathrm{eff}}$, reducing the amplification predicted by Theorem 1. Our probe estimators (App. D.1) explicitly quantify on/off–diagonal kernel scales; deviations can be addressed by whitening probes or by reporting the empirical $d_{\mathrm{eff}}$ in lieu of nominal latent dimension. *Effective dimension.* If model design or decoding choices suppress stochasticity (e.g., deterministic/greedy decoding, very low sampling temperatures), then $d_{\mathrm{eff}}$ may plateau even as the nominal dimension grows, limiting the decay of $\mu_{\mathrm{eff}}$ with dimension. Conversely, architectures that spread sensitivity across layers/time (e.g., diffusion samplers with multiple Gaussian injections) increase $d_{\mathrm{eff}}$.

#### D.3.2 ON THE LOOSENESS OF BOUNDS

Theorem 1 provides an *upper envelope* on $\mu_{\mathrm{eff}}$; empirical tradeoffs can be strictly stronger. Several factors contribute to conservatism: (i) constants hidden in $\lesssim$ (from Cauchy–Schwarz, union bounds, and stability recursions) are not optimized; (ii) the Gaussianization term $\gamma_d$ and the equal-variance slack $\omega$ are upper-bounded via generic inequalities, whereas empirical estimates (App. D.1) are often much smaller; (iii) the score domination step $K_{s,T} \preceq c_s^2 K_{\ell,T}$ introduces a factor $c_s$ that depends on the chosen calibration loss and can be pessimistic if the downstream score $s$ is less sensitive than $\ell_{\mathrm{cal}}$. These choices keep the guarantees uniform across scores but may overestimate the worst-case test power.

#### D.3.3 APPLICABILITY TO OTHER DP MECHANISMS

Our path-kernel view targets *function-level stability* and therefore extends beyond DP–SGD when the mechanism admits a stability certificate that can be mapped to an RKHS norm over probes. *PATE.* Teacher aggregation with Gaussian/Laplace noise yields a score-level sensitivity bound; one can define an output-path kernel from the gradient of the (smoothed) voting score with respect to latent probes and obtain a GDP envelope by repeating our reduction. *CDP/zCDP and Gaussian mechanisms.* When training or release is directly privatized by Gaussian noise in the output channel, the LPK energy $C_T$ is explicit and the same Gaussianization step applies; the envelope recovers the classical Gaussian DP curve with dimension-aware dilution through $d_{\mathrm{eff}}$. *DP variants with momentum/Adam, SGLD.* Stability recursions change but still control a path energy (now including velocity terms); replacing Proposition 1 by the corresponding stability inequality yields the same RKHS-to-GDP transfer. Formalizing these variants is a promising direction for future work.

#### D.3.4 MORE COMPLEX THREAT MODELS

Our analysis is *black-box*: the attacker does not control or observe latent randomness. If the adversary can fix seeds or access internal randomness (gray-box), the Gaussianization step weakens; in the extreme, deterministic decoding collapses $d_{\mathrm{eff}}$ and removes dilution. Adaptive multi-release attacks that correlate queries across time can increase the product-level slack beyond the i.i.d. bound in §4.5. Extending the envelope to such adaptive or side-informed settings is an important direction for future work.

