# OpenReview forum: "Beyond Worst-Case: Dimension-Aware Privacy for Black-Box Generative Models"
_ICLR.cc/2026/Conference — ICLR 2026 Conference Withdrawn Submission_

### Official Review · Reviewer_Kr3F · 2025-10-15

**Soundness:** 3
**Presentation:** 2
**Contribution:** 2
**Rating:** 4
**Confidence:** 5

**Summary:**

This paper presents a theoretical analysis of the black-box privacy of synthetic data generated by differentially private (DP) generative models.
Using tools such as the Loss Path Kernel (LPK) and Gaussianization, it shows that the observable distinguishability between outputs of two neighboring datasets, termed effective distinguishability, scales as $1 / (n \sqrt{d_{\text{eff}}})$, where $n$ is the dataset size and $d_{\text{eff}}$ is the effective latent dimension. The analysis formalizes why synthetic data from DP-trained generators often appear more private than what the DP-SGD upper bound suggests.

**Strengths:**

- Solid theoretical analysis and novel tools: The paper provides a mathematically rigorous treatment of the privacy loss of DP synthetic data.
It introduces novel tools such as LPK and Gaussianization, offering a fresh theoretical lens on DP-SGD
- It tackles an important question and sheds light on why outputs from DP generative models often appear more private than their theoretical guarantees suggest

**Weaknesses:**

- Unclear writing and framing. As a reader, I had to infer both the intent and the main focus throughout. The paper does not clearly state upfront that it analyzes the privacy of synthetic data generated by DP generative models; the title suggests a study of the privacy of DP generative models themselves. There are numerous ambiguous terminology (e.g., “test-based”, “dimension-aware”), which are introduced without sufficient context or motivation.

- The paper fails to make clear that DP-generated synthetic data are expected to be more private than what DP-SGD’s theoretical guarantees suggest, simply because of different adversary assumptions. DP-SGD provides a worst-case guarantee under an adversary who observes all intermediate model states; when one only observes the final iterate or its post-processed outputs (synthetic data), a privacy amplification effect naturally arises. This phenomenon has been widely studied in the literature (see Sec. 5.1 in [1], among others). Thus, the paper mainly offers an alternative mathematical formalization of this predictable result rather than revealing a fundamentally new privacy mechanism. This central point is never made explicit, which risks misleading readers about the work’s conceptual novelty.

- The results hinge critically on the presence of a high-dimensional latent Gaussian variable fed into the generator during inference. While this holds for VAEs or diffusion models, it does not apply to most language models, making the title’s general claim overstated.

- It remains unclear how practitioners could apply these theoretical findings to guide “model selection, hyperparameter tuning, and risk assessment in realistic pipelines” (Sec. 6). The distinguishability results rely on assumptions that might not hold in practice and therefore cannot be treated as formal DP guarantees for the synthetic data. Consequently, the method acts more like a heuristic privacy estimation, similar in spirit to [2]. I expect a detailed discussion and comparison with [2].

- Finally, the paper overlooks a large body work on DP synthetic data, in particular the survey [3] as well as many recent works on DP synthetic text [4,5].

[1] Hu, Yuzheng, et al. "Empirical Privacy Variance." ICML 2025

[2] Steinke, Thomas, et al. "The last iterate advantage: Empirical auditing and principled heuristic analysis of differentially private sgd." ICLR 2025

[3] Hu, Yuzheng, et al. "Sok: Privacy-preserving data synthesis." IEEE S&P 2024.

[4] Yue, Xiang, et al. "Synthetic text generation with differential privacy: A simple and practical recipe." ACL 2023

[5] Tan, Bowen, et al. "Synthesizing privacy-preserving text data via finetuning without finetuning billion-scale llms." ICML 2025

**Questions:**

I don’t have further questions at this point. My main concerns lie in the insufficient contextualization of related literature and the limited practical value of the analysis. I would be happy to reconsider my rating if these issues are adequately addressed.

---

### Official Review · Reviewer_n5E3 · 2025-10-29

**Soundness:** 2
**Presentation:** 3
**Contribution:** 2
**Rating:** 2
**Confidence:** 4

**Summary:**

This paper proposes an interface-specific ROC/GDP upper envelope for membership inference attack and tries it to training stability via a loss-path kernel (LPK). It upper-bounds attack power for a fixed score and latent distribution by integrating (i) LPK-based stability term from the DP-SGD with (ii) a Gaussianization equal-variance reduction that introduces explicit slack terms. The certificate is computed post-hoc from a single training run. The contribution is positiuoned as an operational auditing tool, which is interface-specific and does not change the formal $(\epsilon,\delta)$ DP budget.

**Strengths:**

Combining stability analysis with a Gaussianization and equal-variance reduction and a simple $\sqrt{m}$ use-times composition is very interesting and creative that may remove the need for shadow models or repeated re-training. As an operational auditing tool, I think the method is attractive. The method gives practitioners a post-hoc certificate for a fixed interface.

The paper is generally well structured and the presentation is clear.

**Weaknesses:**

# C1:


The paper's claim to "address the gap ... beyond worst-case" is not supported. The paper does not define or reduce any mechanism-level gap. It presents an interface-specific ROC/GDP upper bound that can sit below the worst-case accountant curve. This is expected and does not isolate a black-box cause. There is no white-box counterfactual, no adaptive-attack analysis, and no evidence that latent randomness explains the discrepancy. The title "Beyond worst-case" is misleading. I think there is nothing surpasses the wrost-case guarantee. Rather, the work specializes it to a fixed interface.




# C2:

The black-box positioning seems misleading and the setting needs more clarification.

The technical pipeline "LPK stability --> Gaussianization/equal-variance--> envelope" does not hinge on "black-box" per se. If we fix the same released score and the same latent distribution, the very same bound would apply whether the attacker is black- or white- box. The math is interface-specific, not observability-specific.


The actually black-box here is as follows. Only the threat model that is evaluated against (attacker restricted to that fixed score and unable to change the latents).

The bound targets black-box distinguishability, but it is not black-box computable. The envelope depends on the loss-path kernel (i.e., gradients along the training trajectory). An external auditor with only API access cannot reconstruct this. So, the result is, at best, an internal, white-box audit about a black-box attacker.


If the contribution were truly about the black-box setting, we would expect either a black-box computable certificate or a clear statement that such a certificate is impossible and thus why a white-box internal one is the best you can do. I think the paper does neither.

In addition, a third party cannot validate the envelope, right? So, only the model owner (with logs/gradients) can. That undermines "auditability" as a general benefit.



# C3:

This paper emphasized LPK as the right geometry. However, it seems that the value of LPK is not established. There is no theoretical analysis showing LPK yeilds a uniformly tighter ROC/GDP envelope than standard baselines. Also, there is no head-to-head empirical ablations on the same runs. Similarly, there is no systematic comparison to the accountant-->GDP curve to show consistent tightening.

Given LPK's extra white-box gradient passes and reliance on near-isotropy/smoothness, the paper needs either a theoretical domiance result or clear empirically better result to justify LPK over the relatively simpler alternatives.


# C4:

The envelope relies on several assumptions and slack terms (e.g., from Gaussian approximation, and the equal-variance), and an i.i.d., non-adaptive query model.


In Theorem 2, the multi-query guarantee result holds only for $\alpha\in(\Gamma_{m}, 1-\Gamma_{m})$, where the slack $\Gamma_{m}$ is specified by Eq. (5).

**C4-Q1:** For what ranges of m, d_{eff}, n (and measured slack constants) is this interval non-empty and practically useful? In particular, if $\Gamma_{m}\geq 0.5$, the result is vacuous. Could you provide a non-vacuity condition observed in your experiments?


**C4-Q2:** Over the valid FPR range, when does $G_{\mu_{eff}}(\alpha, \Gamma_{m})$ sit strictly below the trivial ceiling? If not, the envelope is practically vacuous.


# C5:

The paper's multi-release/query analysis in Theorem 2 relies on an i.i.d. query model were latent variables are drawn independently from a fixed Gaussian distribution.

**What is okay:** The mathematical derivation of the $\sqrt{m}$ composition under this specific i.i.d. sampling process is consistent.


**What is the problem:** This constitutes a significant relaxation of the standard DP threat model. In standard DP composition, the privacy guarantee holds against any **adaptive attacker** who can choose each query based on the results of all previous ones. Here, the i.i.d. assumption explicitly restricts the attacker from this adaptive strategy. A real-world auditor (or attacker) is not limited to random Gaussian probes. They can and will adpat their queries to maximize distinguishability for accurate privacy audition (auditor) or privacy attack (attacker).

As a result, the envelope derived under the assumption of "i.i.d. queries" does not address the "worst-case scenario" it claims to, but rather a much more restricted one. This fundamentally undermines the paper's framing as explaining the robust empirical privacy observed in practice, as it does not account for the worst-case threat models.

**Questions:**

Please see the comments in Weakness.

---

### Official Review · Reviewer_p9xr · 2025-11-02

**Soundness:** 3
**Presentation:** 3
**Contribution:** 3
**Rating:** 6
**Confidence:** 3

**Summary:**

This paper analyzes the observation that black-box evaluations of DP-trained generative models often exhibit stronger privacy than what is predicted by worst-case accounting. The study explains this gap through an f-DP / Gaussian DP testing framework, measuring model stability in the LPK geometry, and introducing dimension-aware GDP envelopes by combining stability with Gaussianization bounds. These envelopes can predict MIA power and thus serve as a quantitative tool for assessing the practical privacy of DP generative models.

**Strengths:**

This study tackles an interesting observation, DP-trained generators appear more private in black-box use.  Two original ideas are proposed to quantify the gap, stability of DP-SGD in the LPK geometry and the dimension-aware GDP envelopes which combine stability with Gaussianization.

The theoretical aspect of the study is strong, showing the formalized stability and the proof of $\sqrt{m}$ composition across multiple releases.

The paper is well-structured and clearly written. Definitions and the reduction steps are explicitly laid out, making the technical pipeline easy to follow.

The empirical results is promosing, whowing the black-box distinguishability decreases with dataset size and effective latent dimension. This offers a helpeful guidance for model selection, tuning, and risk assessment in practice.

**Weaknesses:**

A key weakness is the limited validation, as the experiments involve only a single small VAE on MNIST from one DP-SGD run. How would the proposed envelopes behave on larger architectures or datasets, such as CIFAR-10/100, ImageNet, or diffusion models?

The analysis assumes that a black-box adversary cannot control latent randomness. If an attacker were able to access internal randomness or use deterministic/greedy decoding, how would the Gaussianization effect change? Would the predicted privacy gain become inaccurate or unreliable?

The method relies on assumptions of Gaussian regularity, near-isotropy, and Lipschitz continuity of loss gradients. These assumptions may not hold in the presence of nonsmooth activations (e.g., ReLU) or anisotropic structures. Although workarounds such as smoothing are mentioned, it remains unclear how effective they are in practice.

As noted in Section D.3.2, the envelopes are upper bounds and can be pessimistic, potentially overestimating the worst-case test power. Would it be possible to calibrate these bounds using empirical tradeoff curves, tighter Gaussianization methods, or more realistic loss/score pairings?

Finally, computational cost is an important aspect that should be expanded. Section D.1 states that the overhead is typically less than 10% of wall-clock time with M≤64. However, this evaluation is limited to a single small setup and should be validated on larger models and datasets to confirm its practical usability.

**Questions:**

See above.

---

### Note · Authors · 2025-11-12

I have read and agree with the venue's withdrawal policy on behalf of myself and my co-authors.